# Baltic Sea Surface Temperature Analysis 2022: A Study of Marine Heatwaves and Overall High Seasonal Temperatures

Anja Lindenthal[1] and Claudia Hinrichs[1], Simon Jandt-Scheelke[1], Tim Kruschke[1], Priidik Lagemaa[3], Eefke M. van der Lee[2], Ilja Maljutenko[3], Helen E. Morrison[2], Tabea R. Panteleit[1], Urmas Raudsepp[3]

[1]Federal Maritime and Hydrographic Agency, Hamburg, 20539, Germany
[2]Federal Maritime and Hydrographic Agency, Rostock, 18057, Germany
[3]Department of Marine Systems, Tallinn University of Technology, Tallinn, 12618, Estonia

*Correspondence to*: Anja Lindenthal (anja.lindenthal@bsh.de); Claudia Hinrichs (claudia.hinrichs@bsh.de)

**Abstract.** In 2022, large parts of the Baltic Sea surface experienced the third-warmest to the warmest temperatures over the summer and autumn months since 1997. Warm temperature anomalies can lead to marine heatwaves (MHWs), which are discrete periods of anomalous high temperatures relative to the usual local conditions. Here, we describe the overall sea surface temperature (SST) conditions observed in the Baltic Sea in 2022 and provide a spatio-temporal description of surface MHW events based on remote sensing, reanalysis and in-situ station data. The most MHWs, locally up to seven MHW events, were detected in the western Baltic Sea and the Inner Danish Straits, where maximum MHW intensities reached values of up to 4.6 °C above the climatological mean. The northern Baltic Proper and the Gulf of Bothnia were impacted mainly by two MHWs at maximum intensities of 7.3 °C and 9.6 °C, respectively. Our results also reveal that MHWs in the upper layer occur at a different period than at the bottom layers and are likely driven by different mechanisms. Reanalysis data from two exemplary stations, 'Lighthouse Kiel (LT Kiel)' and 'Northern Baltic', show a significant increase in MHW occurrences, of +0.73 MHW events per decade at LT Kiel and of +0.64 MHW events per decade at Northern Baltic, between 1993 and 2022. Moreover, we discuss the expected future increased occurrence of MHWs based on a statistical analysis at both locations.

## 1 Introduction

Global warming has led to an increase of ocean heat content (OHC) by about 350 ZJ in the upper 2000 meters from 1958 to 2019, with the year 2022 being the warmest on record as of this writing (Cheng et al., 2022; WMO, 2023). Simultaneously, marine heatwaves (MHWs), extreme events of high water temperature (Hobday et al., 2016), have increased in frequency, duration, spatial extent and intensity during the past four decades (Sun et al., 2023). In 2022, MHWs were recorded on 58 % of the ocean surface (WMO, 2023).

The Baltic Sea is one of the marine ecosystems with the fastest recorded warming of surface temperatures of 1.35 °C between 1982 and 2006, i.e., 0.54 °C per decade (Belkin, 2009). SST data operationally produced by the German Federal Maritime and Hydrographic Agency (in the following BSH, product ref. no. 1 in Table 1) show a warming trend of 0.58 °C per decade for

the period 1990–2022. High SSTs can affect phytoplankton production, while unprecedented high temperatures in the subsurface layers of the sea could have even more devastating effects on the marine ecosystem (Kauppi et al., 2023). Conditions that facilitate the fast warming of the Baltic Sea are the limited exchange between surface and deeper layers due to a permanent halocline at a depth of 60–80 m (Väli et al., 2013) and the limited water exchange between the Baltic Sea and the open ocean through the narrow Skagerrak. That is why local air-sea heat exchange is the main physical factor for the surface layer water temperature and heat content in the Baltic Sea (Raudsepp et al., 2022).

Global mean air temperature in 2022 was among the six warmest in the 173-year instrumental record (WMO, 2023). For Europe especially, the Copernicus Climate Change Service/ECMWF (2022a) states that the air temperatures in August 2022 were higher than the 1991–2020 average across most of the continent, especially in a band in Eastern Europe stretching from the Barents and Kara seas to the Caucasus. In November 2022, air temperatures were higher than the 1991–2020 average, especially over the west, south-east and far north of Europe, and were unusually mild over the northern European seas (Copernicus Climate Change Service/ECMWF, 2022b). These large-scale weather patterns likely lead to high sea surface temperatures (SST) in marginal seas like the Baltic Sea and are a likely driver of MHWs. This hypothesis is further supported by a study by Holbrook et al. (2019), which found that MHWs at middle and high latitude regions were driven by large-scale atmospheric pressure anomalies which cause anomalous ocean warming. Stalled atmospheric high-pressure systems coincide with clear skies, warm air, and reduced wind speeds. These conditions then lead to quick warming of the upper ocean and increased thermal stratification due to reduced vertical mixing.

So far, there generally have been only a few studies on MHWs in the Baltic Sea (Goebeler at al., 2022; She et al., 2020). In this study, we show that remote sensing data revealed several SST anomalies over the entire Baltic Sea in 2022. We thus use reanalysis and in-situ station data to provide a spatio-temporal description of the corresponding MHWs. Both datasets contain data collected over a long enough period to also provide its own respective climatology, thereby enabling a consistent representation of MHWs. While the in-situ data provides accurate point-wise measurements of the temperature at selected locations, the reanalysis data allows for a widespread analysis of MHWs over the entire Baltic Sea, including their extension into subsurface layers. Furthermore, we extend our study by providing a climatology of MHWs at two specific mooring locations, namely at the Lighthouse Kiel (LT Kiel) and Northern Baltic stations. The overall aim of this study is to highlight the areas of the Baltic Sea that were (most) affected by MHWs and determine whether surface MHWs can propagate into deeper layers and thus potentially threaten the subsurface ecosystem. Furthermore, analyzing the climatology of MHWs can provide insight into whether the global increase in MHWs can also be expected to occur on a local scale for the Baltic Sea.

## 2 Data and Methods

### 2.1 Satellite data

The satellite data service at the BSH compiles daily maps of SST data (product ref. no. 1 in Table 1). These have contributed, for example, to the assessment of climate change in the Baltic Sea (The BACC Author Team, 2008) and to the model evaluation in the Baltic Sea Model Intercomparison Project (Gröger et al., 2022). The SST data are recorded as radiances by the Advanced Very High Resolution Radiometer (AVHRR/3) in two thermal infrared channels aboard the NOAA-19 and MetOp B satellites, providing a spatial resolution of 1.1 km, swath widths of 1,447 km and orbital periods of 100 minutes (EUMETSAT, 2015; Minnett et al., 2019). The raw data of eight or nine daytime passes over the Baltic and North Sea are received directly from EUMETSAT and processed using automated, standardized correction procedures (atmospheric correction, cloud masking, georeferencing etc.). Additionally, each flyover is corrected manually in order to preserve as much data as possible whilst eliminating any faulty or cloudy pixels. All available single images from a calendar day are combined and averaged, on a single pixel basis, into one daily-mean image. These daily images are then used to produce a weekly analysis on an operational basis. While the BSH has been carrying out the processing of the satellite data itself on the 1.1 km grid since 1990, operational SST analysis for the Baltic Sea did not start until the autumn of 1996. The analysis of the BSH SST dataset presented in this chapter is therefore limited to the period from 1997–2022.

### 2.2 Station data

In-situ temperature time series from mooring stations located in the Baltic Sea are used for 1) model validation and 2) cross validation of the MHW computation from the reanalysis data. Except for SST data from Northern Baltic (K. Hedi, FMI, pers. communication), the station data are obtained from product ref. no. 2 in Table 1. Each available dataset has already been quality controlled by the regional production units (In Situ TAC partners, 2022). The temporal resolution varies from hourly at the German stations to half-hourly at the stations in the northern Baltic Proper and Gulf of Finland. Due to failures, maintenance and other circumstances, no mooring station entirely covers the period from 1st Jan 1993 until now.

Of all available mooring stations, we selected those which contain data from 2022 and from at least ten additional years from 1993 until 2021 at least one depth. Out of the remaining seven mooring stations that contained surface temperature data, two mooring stations were chosen for the cross validation of MHWs: Lighthouse Kiel (LT Kiel) and Northern Baltic (Fig. 1). Regarding the observation data, LT Kiel has the greatest time coverage (1989 until the present, missing data: 9.1 % of days). This mooring station lies in the far western part of the southern Baltic, and the water depth there is about 12 m. The station Northern Baltic is located in the northern Baltic Proper where the SST observations there cover the period from 1997 until now (missing data: 8.0 % of days). No mooring station provides a time series in deeper layers long or consistent enough to analyze subsurface MHWs, thus reducing the scope of measurement-based analysis of MHWs to the surface layers.

## 2.3 Baltic Sea physics reanalysis data

The Baltic Sea physics reanalysis multi-year product (BAL-MYP; product ref. no. 3 in Table 1) is a dataset based on the ocean model NEMO v4.0 (Gurvan et al., 2019). The model system assimilates satellite observations of SST (EU Copernicus Marine Service Product, 2022b) and in-situ temperature and salinity profile observations from the ICES database (ICES Bottle and low-resolution CTD dataset, 2022). The product provides gridded information on SST and subsurface temperature conditions. The spatial coverage is 1 nautical mile, i.e., approximately 1.8 km. The grid covers the entire Baltic Sea, including the transition zone to the North Sea, with a vertical resolution of 56 non-equidistant depth levels. This multi-year product (MYP) covers the reference period from 1993 up to 2022. The model setup is described in the Product User Manual (PUM, Ringgaard et al., 2023).

## 2.4 Heat wave detection

Marine heatwaves refer to a discrete period of unusually high seawater temperatures. While several definitions describe MHWs quantitatively, the most commonly used method defines them as periods when temperatures exceed the 90th percentile of the local climatology for five days or more (Hobday et al. 2016). We use open-source tools to detect MHWs (Oliver, 2016; Zhao and Marin, 2019) in station and reanalysis data. The identified MHWs can be classified following Hobday et al. (2018), in which the MHW category is based on the maximum intensity in multiples of threshold exceedances, i.e., the local difference between the 90th percentile threshold and the climatology: If the threshold is exceeded less than 2 times, the MHW is classified as moderate (Category I), at 2 to 3 times it is classified as strong (Category II), at 3 to 4 times it is classified as severe (Category III), and at 4 or more times it is classified as extreme (Category IV).

Here, the occurrence of MHWs in the Baltic Sea in 2022 is analyzed based on the BAL-MYP (product ref. no 3 in Table 1). MHWs are computed at every third surface grid point, resulting in a resolution of approximately 5.4 km for the following statistical metrics: cumulative intensity, mean intensity, duration of the longest heatwave, number of heatwaves (frequency), maximum intensity and total days of MHW conditions.

Then, in order to evaluate the development of those MHW metrics over time, block averages (using a block length of one year) for each MHW metric are computed for both the observations (product ref. no 2 in Table 1) and the BAL-MYP (product ref. no 3 in Table 1) at two stations: Lighthouse Kiel and Northern Baltic. The yearly MHW metrics from observations and the reanalysis are correlated for evaluation, and linear trends (95 % significance) are calculated for each of those metrics. Finally, the correlation of the annual MHW metrics to the annual mean temperature based on reanalysis data was assessed using a linear least-squares regression and a two-sided t-test for significance.

All MHW assessments in the following sections use the period from 1993 to 2021 for the climatology, except for Sect. 3.2.1, in which the comparison of the multi-year evolution of MHWs at Northern Baltic uses the overlapping period from 1997 to 2021 due to the lack of observations at this station before 1997.

## 2.5 Validation of the Baltic Sea physics reanalysis

The BAL-MYP (product ref. no 3 in Table 1) has already been extensively validated in the corresponding Quality Information Document (QuID; Panteleit et al., 2023), where the reanalysis data is validated within the time period from 1st January 1993 to 31st December 2018. The validation in the QuID shows a negative bias at the surface with a shift towards more positive values at deeper levels. A variation of statistical values with depth is also clearly visible in the estimated accuracy number (EAN), which represents the root-mean square difference (RMSD) of a specific depth layer. The RMSD varies between 0.29 °C at 200–400 m over 0.63 °C at the surface to 1.3 °C at 5–30 m depth.

For this study, we additionally evaluated the BAL-MYP data in more detail using a clustering approach, which offers insights into the overall accuracy of the reanalysis by grouping the errors. This clustering procedure employs the K-means algorithm (Raudsepp and Maljutenko, 2022). In this evaluation, all available data within the model's domain and simulation period are considered. A two-dimensional error space (dS, dT) is established using simultaneously measured temperature and salinity values as the foundation for clustering. Here, $dS=(S_{mod}-S_{obs})$ and $dT=(T_{mod}-T_{obs})$ represent the differences between the reanalysis ($S_{mod}$ and $T_{mod}$) and observed ($S_{obs}$ and $T_{obs}$) salinity and temperature, respectively. The dataset employed in this validation study was sourced from the EMODNET dataset compiled by SMHI (product ref. no. 4 in Table 1). It consists of a total of 3,094,089 observations aligning with the simulation period of the BAL-MYP and covering the years 1993 to 2022. A comprehensive explanation of the k-means-method and detailed results describing the accuracy of the BAL-MYP can be found in Appendix A1. The results can be summarized as in that approximately 82 % of all validation points exhibit relatively low temperature bias, STD, and RMSD (Table A1). The surface layer validation shows that less than 10 % of comparison points have significant temperature errors (Figure A1c). Due to the low proportion of these validation points we do not expect a significant impact on the determination of the surface MHWs and their statistics. Below the surface layer, i.e., at depths ranging from 0.5–40 m, up to 25 % of the points correspond to clusters with temperature errors greater than +/- 2.0 °C; in deeper layers, this percentage gets smaller again (Figure A1c). Consequently, we anticipate that the reanalysis data provides sufficiently accurate information for calculating both surface and subsurface MHWs and their statistics for the Baltic Sea.

The BAL-MYP is also validated in terms of how accurately it reproduces the MHWs of 2022 and how well it represents their characteristics during the overlapping time periods of data availability at the two locations (1993–2022 for LT Kiel, and 1997–2022 for Northern Baltic). For this, the reanalysis was compared to the available station data (product ref. no 2 in Table 1 for LT Kiel and K. Hedi, FMI, pers. communication for Northern Baltic) at these locations. Table 2 shows the Pearson correlation coefficients for the MHW metrics in Fig 4 between observational and reanalysis data for the two stations, which show overall good agreement between the two data sets with respect to MHW detection.

We also compared the annual temperature curves resulting from both the reanalysis and the station data at each location (Fig. A2). Overall, the curves show the same progression. The temperature from the BAL-MYP is generally slightly lower, and consequently this results in a slightly lower temperature climatology and threshold (here, the 90th percentile) on which

MHW detection is based. In general, though, the MHWs and their respective intensities and lengths are detected equally in
both the station and reanalysis data.
**3 Results**
**3.1 Sea surface temperature anomalies in satellite data**
In the summer of 2022, large parts of the Baltic Sea featured strong warm anomalies based on the BSH SST analysis (product
ref. no. 1 in Table 1, Fig. 2). The highest values were up to 3 °C above the long-term mean (1997–2021) in the Bothnian Sea
in June and in the Bothnian Bay in July. In August however, these areas were neutral or exhibited cold anomalies, while the
Baltic Proper as well as the Gulf of Finland and the Gulf of Riga showed the warmest anomalies of +1.5 °C to 2.5 °C. At the
beginning of autumn, the Baltic Sea is marked by a substantial east-to-west gradient of SST anomalies due to a series of
upwelling events along its eastern shores. In November, the whole Baltic Sea features strong warm anomalies, again with peak
values above +2 °C around Southern Sweden.
To provide some climatological context for the observed SST anomalies in a straightforward way, we also present maps
ranking the SST anomalies for the summer and autumn months of 2022 against the same months in previous years (right two
columns of Fig. 2). These anomaly rankings provide information on how extreme an anomaly of a given magnitude is. For
every grid point and for each calendar month, the monthly anomalies are ranked by magnitude. The warm anomalies over
large parts of the Baltic Sea during the summer and autumn of 2022 are among the warmest eight on record for the respective
months. In September, coastal upwelling led to cold anomalies along the eastern shores, but the other five months of the
summer and fall of 2022 (June, July and August as well as October and November) show large areas of the Baltic Sea with
warm anomalies that are among the four most pronounced on record. In August and November, we see several large areas
along the coastlines of the Baltic countries as well as off the Polish coast and around Gotland that according to the BSH SST
analysis dataset featured highest-ever surface temperatures.
**3.2 Marine heatwaves**
MHWs describe exceptionally warm temperature anomalies. As the monthly overview in Fig. 2 already provides an indication
of possible MHW conditions in 2022, the MHW metrics defined by Hobday et al. (2016) are assessed using the BAL-MYP
(product ref. no. 3 in Table 1). Each region of the Baltic Sea experienced different MHW characteristics during 2022 (Fig. 3,
Table 3).
The most MHWs during 2022 occurred in the Inner Danish Straits and the Western Baltic (Fig. 3d); mainly, four to five MHWs
were detected, with some assessed locations experiencing up to seven MHWs and a maximum of 94 total days of MHW
conditions (Fig. 3f). The mean and maximum intensities of all MHWs in the Western Baltic reached up to 3.8 °C and 4.6 °C,
respectively (Fig. 3b and 3e). The highest mean and maximum intensity values were reached in the northern Baltic Proper and
in the Bothnian Sea and Bothnian Bay (Fig. 3b and 3e), though these regions were affected mainly by only two MHWs. The
maximum intensity in the Bothnian Bay even reached 9.6 °C, the highest within the entire studied period from 1993 to 2022.
The longest MHW is found in the Baltic Proper (32 days), followed by the Bothnian Sea (31 days) and the Inner Danish Straits
(29 days) (Fig. 3c). The highest values of cumulative intensity (of a single MHW), with up to 119.3 days °C, are found in the
Kvarken, a strait between the Bothnian Sea and the Bothnian Bay (Fig. 3a).

### 3.2.1 Multi-year evaluation of MHW metrics

Next, we assess the frequency and other characteristics of the MHWs that occurred in 2022 in a climatological context based
on both observations and reanalysis data for the two stations, LT Kiel (based on the overlapping climatology period 1993–
2021, Fig. 4a–h) and Northern Baltic (based on the overlapping climatology period 1997–2021, Fig. 4 i–p). Overall, the results
for the yearly MHW metric calculation are well correlated between the observations and the reanalysis data (Table 2).
In 2022, a total of five MHWs (four in the BAL-MYP) occurred throughout the year at LT Kiel (Fig. A2a). Though none of
them was extraordinarily long or intense at LT Kiel, the time series of yearly MHW metrics shows that, based on observational
data, the number of MHW occurrences in 2022 was the second highest there since 1989 (Fig. 4a). The time series of MHW
frequencies per year suggests that the occurrence of MHW events has increased over the last three decades (Fig. 4a). The trend
computed from reanalysis data is +0.73 MHWs per decade for the period 1993–2022. The number of MHW events per year is
positively correlated (R=0.76) with the increasing annual mean SST at this mooring station (Fig. 4b). The maximum (Fig. 4c)
and cumulative intensities (Fig. 4e) of observed MHWs do not show a clear trend and are not correlated to the warming annual
mean temperatures (Fig. 4d and Fig. 4f). There is no significant trend in total MHW days (Fig. 4g) at LT Kiel, but a positive
correlation (R=0.71) with rising average temperatures (Fig. 4h).
For Northern Baltic, neither the station data nor the reanalysis data exhibits a statistically significant trend in MHW events for
the overlapping period (Fig. 4i). But when all of the available reanalysis data from 1993–2022 is taken into account, the trend
in MHW occurrences becomes significant at the 95 % level, with +0.64 MHWs per decade. Again, the number of events is
positively correlated with annual mean temperature (R=0.58, Fig. 4j). The highest maximum MHW intensities were recorded
in recent years (2016, 2018, 2021, 2022), with 2022 showing the highest intensity of any MHW, at 7.3 °C (reanalysis data) to
7.4 °C (station data) above the climatologically expected temperature (Fig. 4k,l, see also Fig. A2b). The cumulative MHW
intensities show no clear trend or correlation with annual mean temperatures at this station (Fig. 4m,n). In terms of total MHW
days, 2018 shows the highest numbers (Fig. 4o), but otherwise no trend is detectable for this metric, though there is positive
correlation with annual mean temperatures (R=0.56, Fig. 4p).

### 3.2.2 Analysis of vertical MHW distribution at Northern Baltic

At Northern Baltic, which is about 103 m deep and located in the Western Baltic Proper, the surface temperature has been
measured continuously over several decades. However, no quality-controlled temperature measurements exist for the lower
layers at this station. The validation of the BAL-MYP shows that, at other locations, the reanalysis represents temperatures
generally well, both at the surface and in the lower stratum. In order to obtain further insights into heat wave propagation
towards the seafloor, we analyzed the reanalysis data along the water column.
A seasonal SST signal is clearly visible in Fig. 5a. In general, the temperature tends to decrease with depth while the bottom
temperature is comparably cold and uniform. In early summer (June), a so-called cold intermediate layer (CIL), defined as a
minimum temperature between the thermocline and the perennial halocline (Chubarenko et al., 2017; Dutheil et al., 2022),
develops at a depth of 20–60 m and acts as a barrier between the surface and bottom water bodies. At Northern Baltic, the
upper boundary of the CIL coincides with the mixed layer depth (MLD), which is depicted in Fig. 5b-c. Starting from around
June, a water stratum with a significantly lower temperature than the climatological mean (up to -7 °C deviation) is located
immediately under the MLD (Fig. 5b), which suggests that the CIL was significantly colder at this time in 2022. This also
coincides with the onset of significantly higher temperatures near the surface, at 0.5 m depth, compared to the climatological
mean, though these were initially not high enough to result in a MHW (Fig. 5e). At this depth, there is a significant temperature
surge of 5 °C above the climatological mean, followed by abrupt and substantial fluctuations in temperature within in a brief
timeframe. This eventually leads to a MHW which lasts for 15 days starting from the end of June and which contains a one-
day extreme MHW (Category IV) event at a temperature of 7.4 °C above the climatological mean, followed by a severe MHW
(Category III) for another three days. Significant temperature deviations can also be observed at a depth of 10.8 m, i.e., at the
MLD, after July 2nd, just after the Category IV MHW at 0.5 m depth. However, these temperature deviations did not result in
a MHW at 10.8 m depth. A comparably weaker MHW can be detected in mid-August at both 0.5 m (Fig. 5e) and 10.8 m
(Fig. 5f). Thus, this weaker MHW penetrates past the MLD into slightly deeper levels before reaching the comparably cold
layer of water underneath.
As shown in Fig. 5c and Fig. 5d, the intensity of the MHW tends to decrease as the depth increases. Four MHWs in regions
close to the seafloor (i.e., below 60 m) were detected during specific periods from February to April, September to October,
and in December. These MHWs are mostly moderate, with temperatures reaching up to 1.59 °C above the climatological mean.
At the end of September, merely three days can be classified as a Category II MHW in one specific depth-layer close to the
seafloor. In the bottom-most depth-layer, the corresponding subsurface MHW is interrupted by five days of temperatures below
the 90th percentile. However, as the temperatures are only slightly below the threshold and the MHW criteria are still met in
the depth-layers above, one might still count this as one continuous MHW. Furthermore, Fig. 5c also shows isolated Category
I MHWs at depths between 20 and 50 m.
**4 Discussion and Conclusions**
During August and November 2022, record-warm sea surface temperatures were observed in substantial areas of the Baltic
Sea proper. Large parts of the Baltic Sea exhibited the third-warmest to the warmest temperatures in summer and autumn

months since 1997. Both periods, in August and November, coincided with atmospheric temperature anomalies. Over the entire year of 2022, the distribution of quantity and intensity of MHWs within the Baltic Sea is twofold: up to seven individual MHW occurrences were recorded as well as simulated in the south-western part of the Baltic Sea, and as a result this region experienced the maximum number of total MHW days of anywhere in the Baltic Sea in 2022. In the northern Baltic Sea, the number of MHWs was lower, with some locations registering only one MHW; remarkably, however, this one MHW led to the highest mean and maximum MHW intensities in the Baltic Sea since the reanalysis started in 1993. In some areas in the Bothnian Bay, the BAL-MYP revealed temperatures that exceeded 9 °C above the 90th percentile of the climatologically expected temperature values (Fig. 3d,e). This can be considered an extraordinarily high MHW intensity, since maximum SST anomalies above 5 °C have only been observed in about 5 % of the global ocean, and MHW intensities normally peak at 2.5 °C to 3.7 °C (Sen Gupta et al., 2020). In our case, the area in the Bothnian Bay experienced a short period with southerly winds and air temperatures up to 28 °C at the end of June 2022 (SMHI, 2023), which led to a short, but very intense MHW in the shallow areas of the Bay.

A significant increase in MHW occurrences is detectable over time at our two exemplary stations, of +0.73 MHW events per decade at LT Kiel and +0.64 MHW events per decade at Northern Baltic. Both MHW frequency and the total number of MHW days are statistically related to rising mean temperatures. This confirms that an increasing number of MHWs can be expected in the future in the Baltic Sea, too, due to global warming (Frölicher et al., 2018; Oliver et al., 2019). The adverse impact of MHWs on the ecosystem's various trophic levels has been widely documented (Smale et al., 2019; IPCC, 2022; Smith et al., 2023). The Baltic Sea, which has a relatively vulnerable ecosystem, could experience a significant negative impact from MHWs (Kauppi and Villnäs, 2022; Kauppi et al., 2023), and the analysis of subsurface MHWs opens up further potential ways to study their effects. At the Northern Baltic mooring station, MHWs were found close to the surface, propagating into deeper layers until reaching the CIL, and some were also detected close to the seafloor. Isolated MHWs were also observed at depths between 20 and 50 m. However, these are subject to higher uncertainty compared to the ones in the surface and bottom layers due to a higher uncertainty in modeling variability in the pycnocline (QuID; Panteleit et al., 2023). Possible reasons for the development of the four MHWs close to the seafloor at Northern Baltic could, for example, be vertical heat transport from the surface or a lateral transport of warmer water due to bottom currents. However, a more detailed evaluation would be required to assess their precise cause.

Potential avenues for future studies include examining whether and how surface MHWs are able to propagate into the deeper water masses close to the halocline as well as examining the correlation between the strength (i.e., the classification category) of the MHW and its propagation into deeper water masses. At Northern Baltic, severe and extreme MHWs occurred close to the surface when the CIL was particularly cold compared to the climatology. This therefore raises questions of whether a strong CIL might be linked to the development of MHWs at the surface and whether the one might even favor the development of the other. Additional studies could also focus on the positive feedback on the bottom temperature, as was observed in 2022. It might be interesting to determine if this phenomenon can also be found in other years and whether it is triggered by the

superposition of either lateral currents or MHWs or of both together. Understanding the effects that potentially lead to the
vertical propagation of MHWs like those observed particularly in the late summer of 2022 will become increasingly crucial in
order to evaluate how the already-increasing occurrences of surface MHWs may affect the ecosystem in subsurface layers.
**Appendix A1**
We apply a clustering approach to evaluate the precision of the Baltic Sea physics reanalysis multi-year product (BAL-MYP,
product ref. no. 3 in Table 1) in order to highlight its ability to accurately capture both surface and subsurface MHWs over the
entire domain. This clustering approach offers insights into the overall accuracy of the reanalysis with respect to temperature
and salinity by grouping the respective errors. The procedure employs the K-means algorithm, a type of unsupervised machine
learning (Jain, 2010). The original explanation of this technique can be found in a study by Raudsepp and Maljutenko (2022).
In our evaluation, all available data within the model's domain and simulation period are considered, even if the observation
data is unevenly distributed or occasionally sparse. This strategy enables us to assess the quality of the reanalysis at each
specific location and time instance at which measurements have been acquired.
Initially, a two-dimensional error space (dS, dT) was established using simultaneously-measured temperature and salinity
values as the foundation for clustering. Here, $dS = (S_{mod} - S_{obs})$ and $dT = (T_{mod} - T_{obs})$ represent the differences between the model
($S_{mod}$ and $T_{mod}$) and observed ($S_{obs}$ and $T_{obs}$) salinity and temperature, respectively. The dataset employed in this validation
study was sourced from the EMODNET dataset compiled by SMHI (product ref. no. 4 in Table 1). It consists of a total of
3,094,089 observations aligning with the simulation period of the BAL-MYP and covering the years 1993 to 2022. For each
observation, we extracted the nearest model values from the reanalysis dataset.
The next stage involves choosing the number of clusters, and for simplicity we opted in advance for five clusters. Subsequently,
the third step entails conducting K-means clustering on the two-dimensional errors. This clustering process is applied to the
normalized errors achieved through separate normalization for temperature and salinity errors using the corresponding standard
deviations. The K-means algorithm then identifies the centroids' positions within the error space for the predetermined number
of clusters. These centroids' locations signify the bias of the error set for each cluster. In the fourth step, statistical metrics for
non-normalized clustered errors are computed. Standard deviation (STD), root mean square deviation (RMSD) and the
correlation coefficient are examples of common statistics that can be calculated for the parameters associated with each cluster.
The fifth step involves examining the spatio-temporal distributions of errors associated with different clusters. During the
creation of the error space, we retained the coordinates of each error point (dS, dT)(x, y), allowing us to map the errors of each
cluster back onto the locations where the measurements were conducted. To achieve this, the model domain is partitioned into
horizontal grid cells (i, j) of $27 \times 27$ km$^2$ in size. Subsequently, the number of error points attributed to various clusters at each
grid cell (i, j) is tallied. The total number of error points linked to the grid cell (i, j) is the sum of points from each cluster. The

proportion of error points in each grid cell affiliated with cluster k is determined by the ratio of the number of error points of cluster k to the total number of error points in each grid cell.

Figure A1 displays the results of the K-means clustering for non-normalized errors. Table S1 presents the corresponding metrics. Within cluster k=5, the salinity and temperature values closely align with the observations, with a bias of dS=-0.40 g/kg and dT=-0.02 °C, respectively. This cluster encompasses 57 % of all data points. The points are distributed throughout the Baltic Sea and the great majority of them exceed 0.5 (Figure A1b). Clusters k=3 and k=4 exhibit relatively even spatial distributions across the Baltic Sea, accounting for 11 % and 8 % of the points, respectively. These clusters are particularly noteworthy due to their relatively high temperature biases and variability, both of which are crucial for the calculation of marine heatwaves. The clusters k=1 and k=2 represent points with low temperature but a high salinity error (Table A1). Spatially, these points are predominantly located in the southwestern Baltic Sea (Figure A1b), which points to the occasional underestimation or overestimation of the inflow/outflow salinity.

Collectively, approximately 82 % of all validation points exhibit relatively low temperature bias, STD and RMSD (Table A1). The surface-layer validation shows that less than 10 % of comparison points have significant temperature errors (Figure A1c). Due to the low proportion of these validation points, we do not expect a significant impact on the determination of surface MHWs and their statistics. Below the surface layer, i.e., at depths ranging from 0.5–40 m, up to 25 % of the points correspond to clusters k=3 and k=4 (Figure A1c). Consequently, we anticipate that the reanalysis data provides sufficiently accurate information for calculating subsurface MHWs and their statistics for the Baltic Sea.

**Data availability**

This study is based on public databases and the references are listed in Table 1.

**Author contribution**

The idea for and concept behind this chapter were formed by Anja Lindenthal, Claudia Hinrichs, Priidik Lagemaa, Helen E. Morrison and Urmas Raudsepp. The data curation was done by Eefke M. van der Lee and Tim Kruschke for the data from product ref. no. 1 in Table 1, by Claudia Hinrichs and Tabea R. Panteleit for the data from product ref. no. 2 in Table 1 and by Simon Jandt-Scheelke and Tabea R. Panteleit for the data from product ref. no. 3 in Table 1. The formal analyses of the datasets and the resulting investigations were performed by Anja Lindenthal, Claudia Hinrichs, Simon Jandt-Scheelke, Tim Kruschke and Tabea R. Panteleit. The k-means model validation was performed by Urmas Raudsepp and Ilja Maljutenko. Claudia Hinrichs, Simon Jandt-Scheelke, Ilja Maljutenko, Tim Kruschke and Tabea R. Panteleit were responsible for the visualization of the data. Anja Lindenthal, Claudia Hinrichs, Simon Jandt-Scheelke, Tim Kruschke, Eefke M. van der Lee, Tabea R. Panteleit

and Urmas Raudsepp were involved in the original draft preparation. The final manuscript was reviewed and edited by Claudia

Hinrichs, Priidik Lagemaa, Helen E. Morrison and Urmas Raudsepp with contributions from all co-authors.

**Competing interests**

The authors declare that they have no conflict of interest.

**Funding**

This work is supported by the Copernicus Marine Service for the Baltic Sea Monitoring and Forecasting Center (21002L2-COP-MFC BAL-5200).

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

    **Tables**


**Table 1: Product Table**

| Product ref. no. | Product ID & type | Data access | Documentation |
|---|---|---|---|
| 1 | BSH Sea Surface Temperature (AVHRR/3); Satellite data | Upon request; overview and contact data via https://www.bsh.de/EN/TOPICS/Monitoring_systems/Remote_sensing/remote_sensing_node.html | https://www.bsh.de/DE/THEMEN/Beobachtungssysteme/Fernerkundung/fernerkundung_node.html |
| 2 | INSITU_GLO_PHYBGCWAV_DISCRETE_MYNRT_013_030; In-Situ Near-Real-Time Observations | EU Copernicus Marine Service Product (2022a) | Quality Information Document (QUID): Wehde et al. (2022) Product User Manual (PUM): In Situ TAC partners (2022) |
| 3 | BALTICSEA_MULTIYEAR_PHY_003_011 (BAL-MYP); Numerical models | EU Copernicus Marine Service Product (2023) | Quality Information Document (QUID): Panteleit et al. (2023) Product User Manual (PUM): Ringgaard et al. (2023) |
| 4 | EMODNET_CHEMISTRY_Baltic_Sea_aggregated_eutrophication_and_acidity_datasets_1902-2017_v2018; Observations | SMHI (2019) | Buga et al. (2018), Giorgetti et al. (2020) |


**Table 2: Pearson correlation coefficients from linear regression between the MHW metrics computed from the station data and the**
**reanalysis data at the stations Lighthouse Kiel and Northern Baltic.**

| Station | common climatology period | MHW count | MHW max intensity | MHW cumulative intensity | total MHW days |
|---|---|---|---|---|---|
| Lighthouse Kiel | 1993-2021 | 0.82 | 0.88 | 0.66 | 0.93 |
| Northern Baltic | 1997-2021 | 0.74 | 0.89 | 0.82 | 0.94 |


**Table 3: Statistical MHW parameter values in various subregions of the Baltic Sea for 2022 based on the reanalysis data from the**
**BAL-MYP (product ref. no. 3 in Table 1) using daily values of SST between 1st January 1993 and 31st December 2022. The**
**climatological period covers the years 1993 to 2021.**

| | Kattegat | Inner Danish Straits | Western Baltic | Baltic Proper | Gulf of Riga | Gulf of Finland | Archipe-lago Sea | Bothnian Sea | Bothnian Bay |
|---|---|---|---|---|---|---|---|---|---|
| Cumulative intensity of longest MHW / °C days | 81.5 | 63.8 | 64 | 79.4 | 63 | 66.5 | 61.1 | 119.3 | 85.1 |
| Mean intensity / °C | 3.6 | 3.5 | 3.8 | 5.3 | 4.9 | 5.8 | 4.5 | 6.4 | 6.5 |
| Duration of longest MHW / days | 24 | 29 | 26 | 32 | 17 | 17 | 21 | 31 | 20 |
| Number of MHWs (modal) per year | 1-6 (3) | 2-7 (4) | 2-7 (5) | 1-7 (3) | 1-4 (3) | 1-4 (2) | 2-4 (3) | 1-6 (2) | 1-5 (2) |
| Maximum intensity / °C | 4.5 | 4.2 | 4.6 | 7.3 | 5.9 | 6.8 | 5.1 | 8.6 | 9.6 |
| Total days of MHW conditions / days | 56 | 86 | 94 | 79 | 50 | 48 | 55 | 63 | 47 |


**Table A1: The share (%), bias, root-mean-square error (RMSE), standard deviation (SD), and correlation coefficient (Corr) for**
**each of the five clusters.**

| | Shares | Bias | | SD | | RMSE | | Corr | | |
|---|---|---|---|---|---|---|---|---|---|---|
| k | % | dS | dT | dS | dT | S | T | S | T | dSdT |
| | | (g/kg) | (°C) | (g/kg) | (°C) | (g/kg) | (°C) | | | |
| 1 | 18.6 | -4.14 | -0.26 | 1.80 | 0.85 | 4.51 | 0.89 | 0.90 | 0.78 | -0.09 |
| 2 | 7.4 | 3.53 | 0.39 | 2.16 | 1.06 | 4.14 | 1.13 | 0.93 | 0.75 | -0.11 |
| 3 | 10.5 | -0.62 | 2.58 | 2.12 | 1.28 | 2.21 | 2.88 | 0.97 | 0.58 | -0.06 |
| 4 | 6.3 | 0.27 | -2.29 | 1.97 | 1.21 | 1.99 | 2.59 | 0.95 | 0.71 | -0.14 |
| 5 | 57.2 | -0.40 | -0.02 | 0.83 | 0.54 | 0.92 | 0.54 | 0.99 | 0.89 | 0.07 |


**Figures**

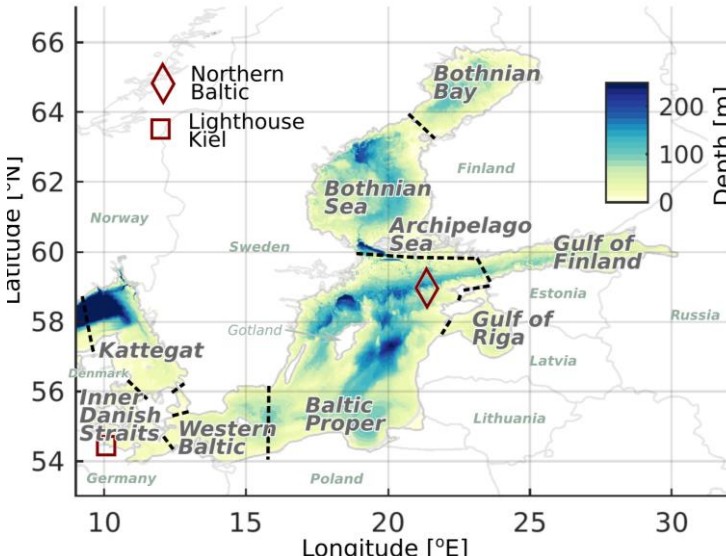


**Figure 1: Map of the Baltic Sea with relevant locations mentioned in the study. Boundaries between subregions are marked with**
**dashed lines.**

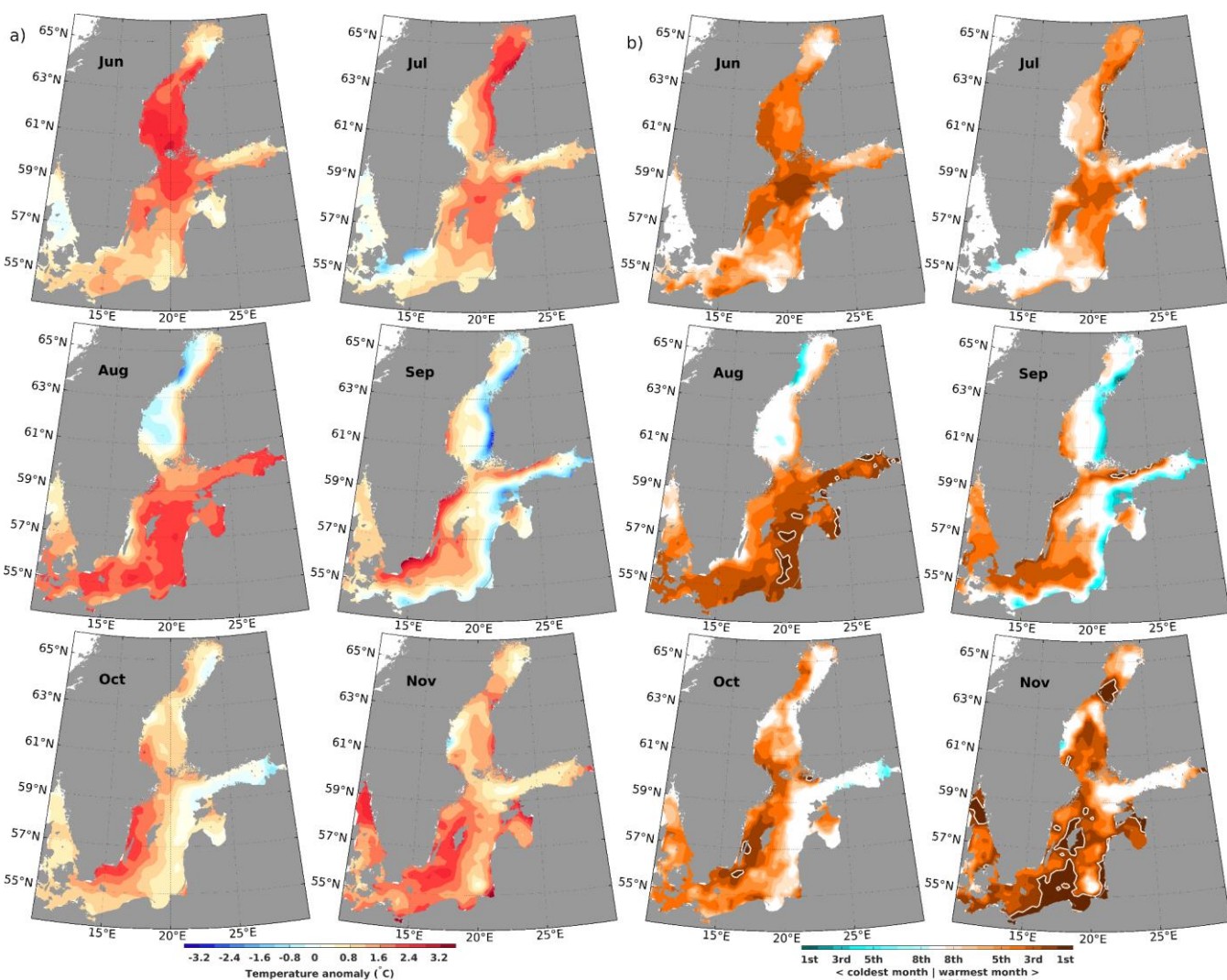


Figure 2: Anomalies (difference to climatology of 1997-2021) of SST for the Baltic Sea according to the BSH SST analysis (product
ref. no 1 in Table 1) during the summer and autumn months in 2022 (a) and ranks of these SST anomalies (b) when compared to the
full dataset starting in 1997. In (b), brownish (cyan) colors denote anomalies belonging to the warmest (coldest) eight anomalies
found since 1997. Record warm anomalies (rank 1) are highlighted by white contours.

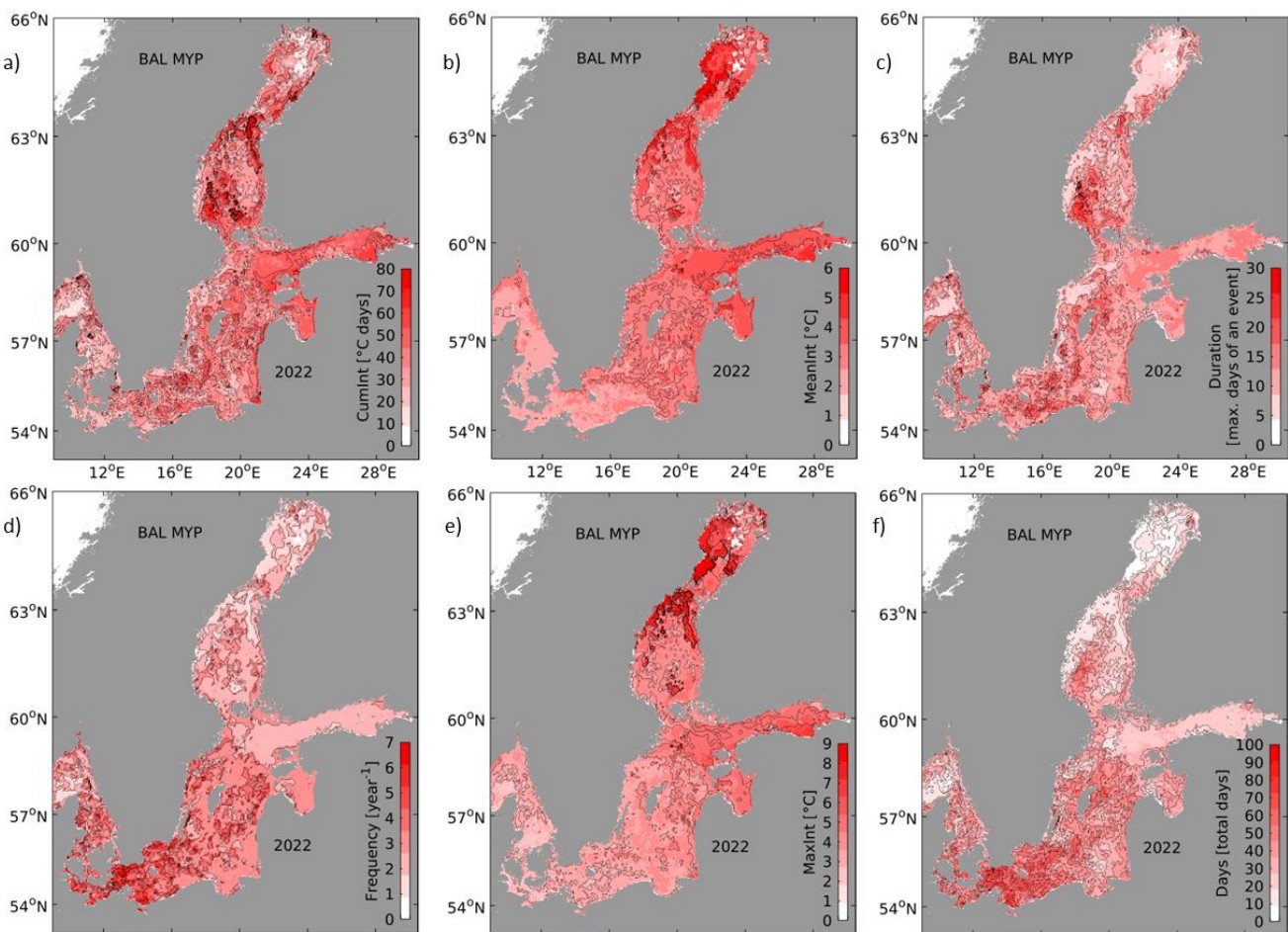


**Figure 3: Statistical metrics of MHWs in 2022 in the Baltic Sea based on SST data of the BAL-MYP (product ref. no. 3 in Table 1)**
**with the climatological period covering the years 1993 to 2021 - (a) cumulative intensity of the longest heatwave, (b) mean intensity,**
**(c) duration of the longest heatwave, (d) number of heatwaves during 2022, (e) maximum intensity during the longest heatwave, (f)**
**summed up days of all heatwave during 2022. The definition of these metrics follows** Hobday et al. (2016).

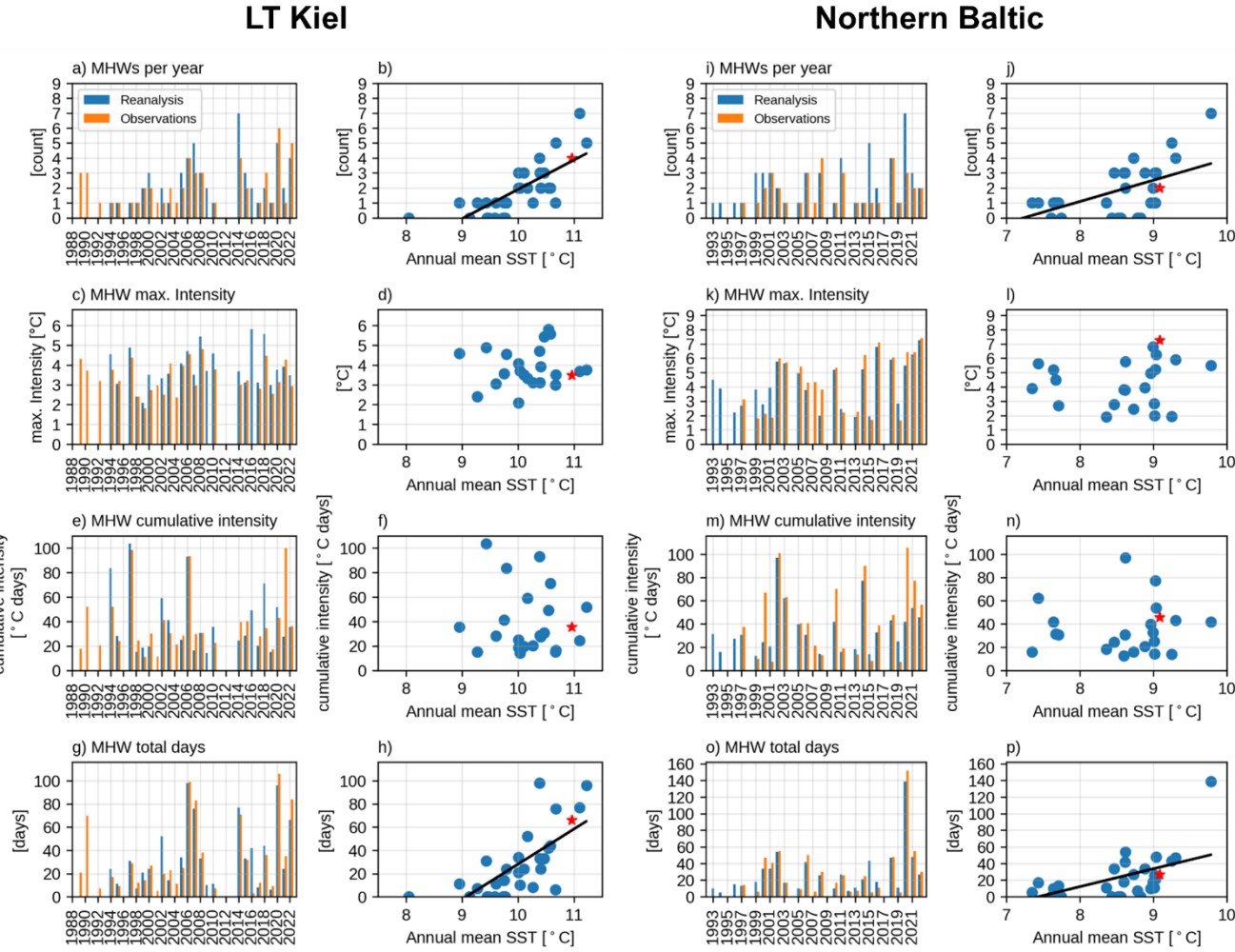

Figure 4: Comparison and time series of annual MHW metrics (a,i: MHW events; c,k: maximum intensity [°C]; e,m: cumulative intensity [°C days]; g,o: MHW days) for station data (orange bars) and BAL-MYP (blue bars) at the stations LT Kiel (left) and Northern Baltic (right). The MHW metrics from the reanalysis are plotted against the annual mean SST at that station with the year 2022 marked in red. Statistically significant (95 %) correlations are indicated with a black line.

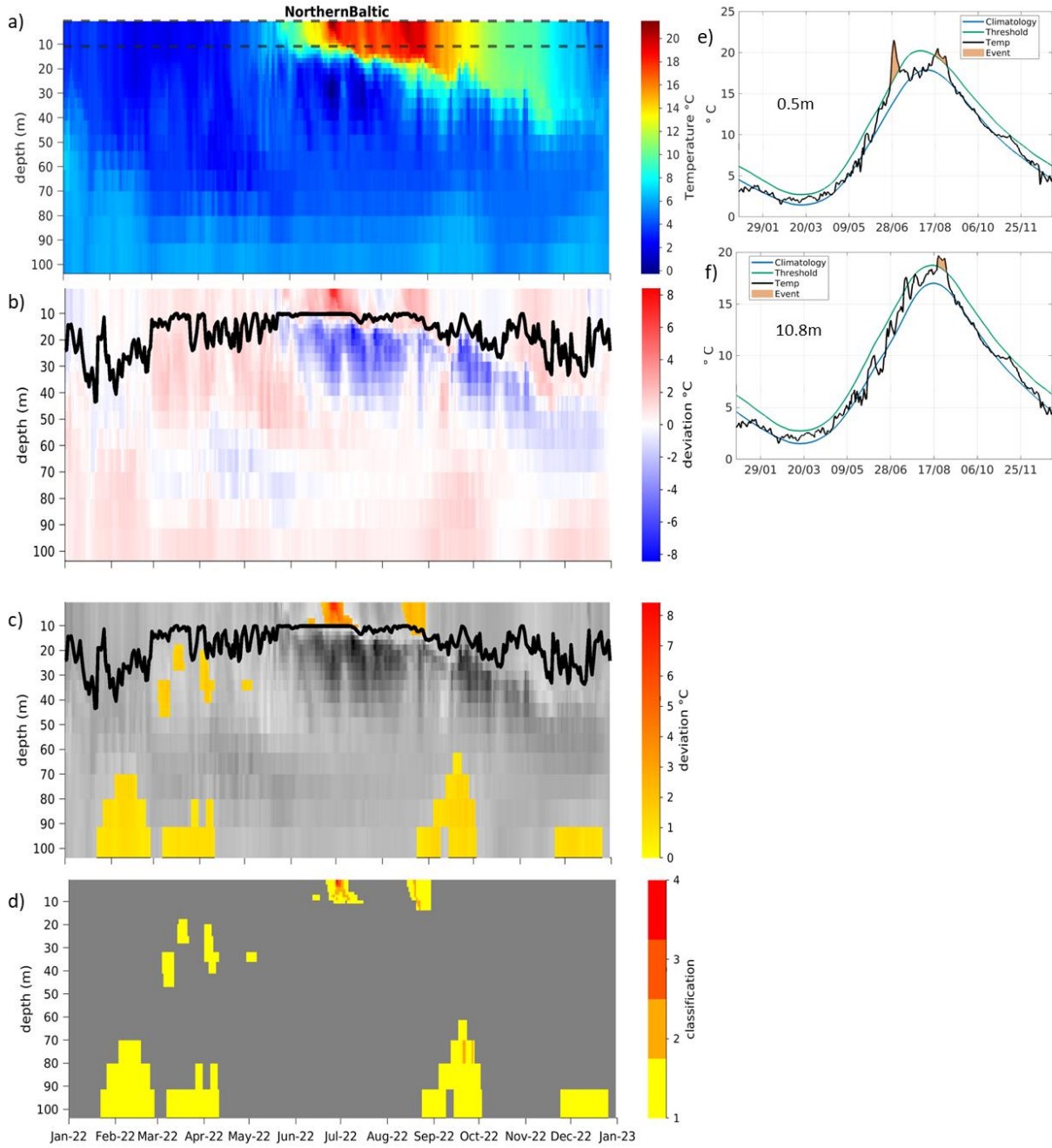


**Figure 5: Hovmöller diagrams show absolute water temperature (a), temperature deviation between the climatology and the BAL-**
**MYP data for 2022 (b) and MHWs (c) and their classifications (d, 1-moderate, 2-strong, 3-severe, 4-extreme) including the mixed**
**layer depth as the thick black line (b and c) at Northern Baltic based on the BAL-MYP (product ref. no. 3 in Table 1). The time**
**series on the right (e-f) are located at the vertical positions marked as dashed lines in (a) and show temperature (black), climatology**
**(blue), 90[th] percentile threshold for MHW analysis (green) and MHWs (red shading) based on reanalysis data at depths of 0.5 m (e)**
**and 10.8 m (f). The period used for the climatology is 1993-2021.**

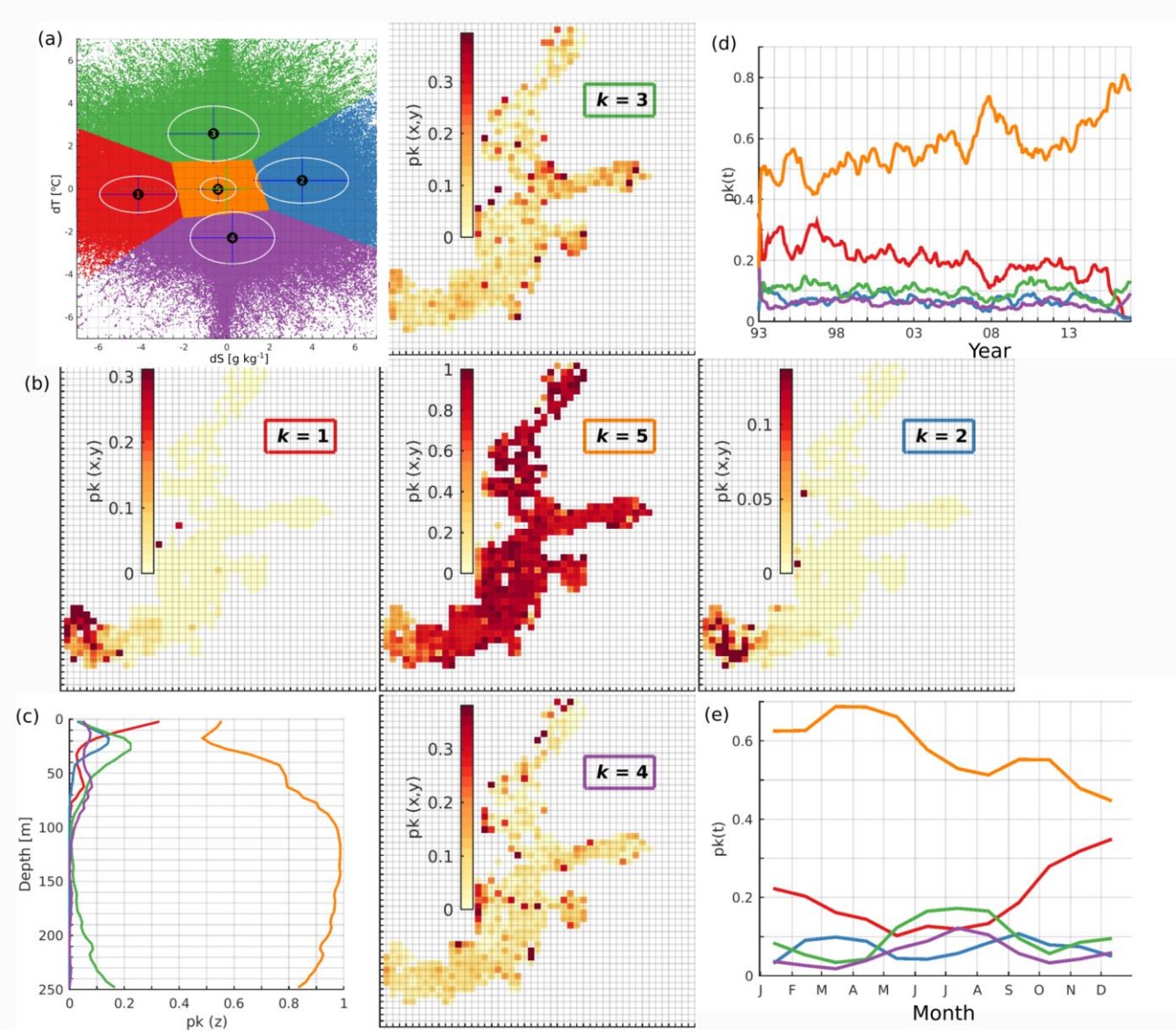


Figure A1: Distribution of normalized error clusters for the BAL-MYP for k=5 (a) and the spatial distribution (b, shaded sub-plots), vertical distribution (c), temporal distribution (d), and seasonal distribution (e) of the share of error points belonging to the five different clusters.

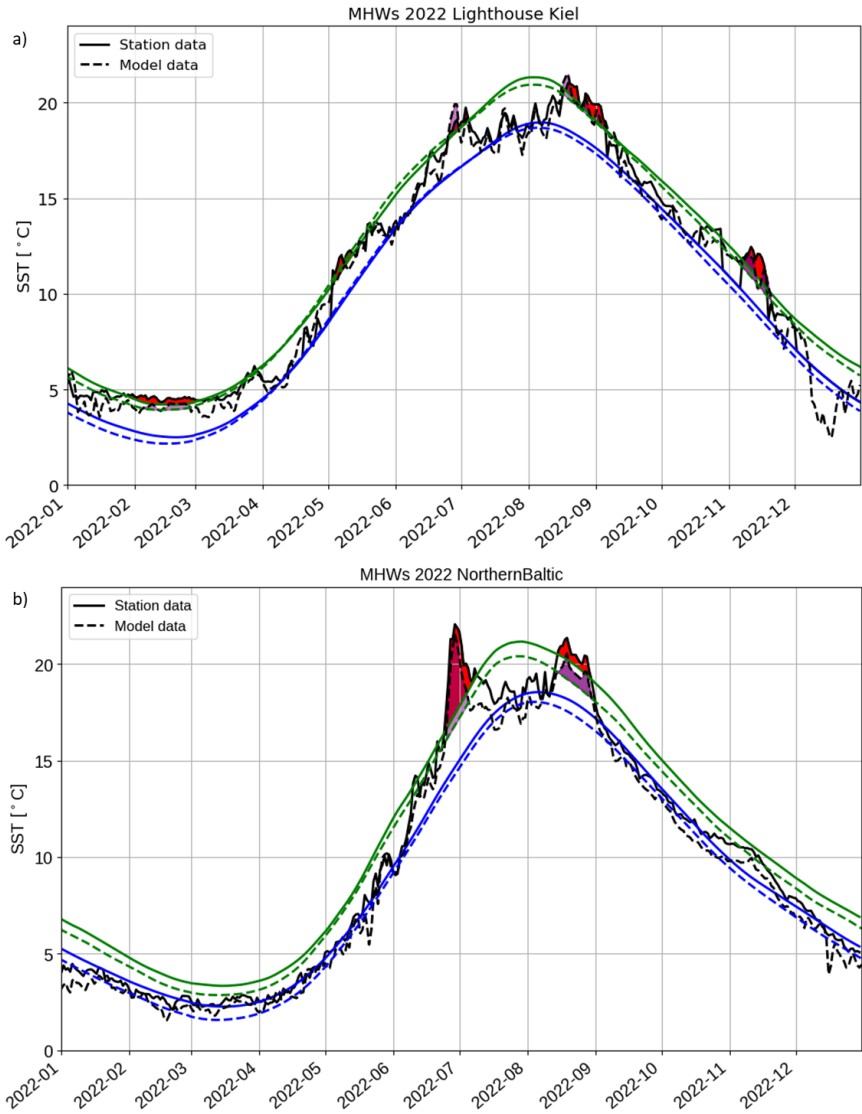


Figure A2: Comparison of station data with BAL-MYP data at (a) LT Kiel (product ref. no. 2 and 3 in Table 1), (b) Northern Baltic
(K. Hedi, FMI, pers. communication and product ref. no. 3 in Table 1). The dashed lines correspond to the reanalysis, while the
continuous lines correspond to the station data. In blue, the climatological mean is shown. The green lines show the 90th percentile
threshold for MHW detection and the black lines are the respective 2022 temperature data. The purple (BAL-MYP) and red (station
data) marked areas show the detected MHWs in 2022. The reference period is 1993-2021 for LT Kiel (a) and 1997-2021 for Northern
Baltic (b).