# Peer review of "Baltic Sea Surface Temperature Analysis 2022: A Study of Marine"

_State of the Planet, 2023_

## Referee Comment (RC2)

**Review of "Baltic Sea Surface Temperature Analysis 2022: A Study of Marine Heatwaves and Overall High Seasonal Temperatures" by Lindenthal et al. for the 8th edition of the Copernicus Marine Service Ocean State Report (OSR 8)**

- ### SUMMARY

The present work conducts an analysis of the Marine Heatwaves (MHWs) detected in the Baltic Sea during the year 2022. To achieve this, the authors utilize various observational and modeling databases. The paper discusses the obtained results, focusing on the analysis of parameters characterizing the variability of MHWs throughout the year (intensity, frequency, and duration). The study describes the magnitude of MHWs in the Baltic environment and draws other interesting conclusions, such as the emergence of positive trends and the relationship between the vertical propagation of MHWs and the development of cold intermediate layers.

Hence, the preprint holds scientific value and falls within the scope of the Ocean State Report. However, some aspects requiring improvement have been identified for manuscript publication. These changes do not alter the substance of the work, allowing its publication with the implementation of minor revisions. Below is a list of the main reasons supporting this recommendation.

As a scientific reviewer, I am compelled to ensure the scientific quality of the contribution. Thus, despite being aware of the space limitations imposed on OSR submissions, some of the recommendations provided may conflict with these constraints. I leave it to the editor's discretion to decide on the implementation of such changes.

- ### GENERAL COMMENTS

The paper employs a considerable amount of geographical terminology (Bothnian Sea, Bothnian Bay, Baltic Proper, Gulf of Finland and Gulf of Riga). The use of this terminology is enriching and aids in the writing process; however, it should be noted that potential readers may not be familiar with Baltic geography. Therefore, I believe that the inclusion of an initial figure (map) displaying relevant data from the study area would be highly beneficial. This map could encompass the following information: Delimitation of the geographical zones used in the article, the location of observational stations (the markers in Figure 2 are hardly visible), bathymetry, etc.

In the Introduction, I miss a clear statement of the objectives and motivation of this work.

One of the weakest points of the paper is related to the model validation (Section 2.4). While the validation exercise is appropriate, it is done inadequately in the paper. Additionally, the graphs used make it difficult to observe the drawn conclusions (see technical corrections). If the aim is to validate the model results regarding the detection of MHWs, I suggest that what should be validated is the model's estimation of the various parameters characterizing MHWs (i.e., the number of events, maximum intensity, cumulative intensity, and MHW days). This comparison can be easily conducted from the data already calculated in Figure 4 and presented to the reader using scatter plots and linear regressions.

The analysis of the vertical structure of MHWs is of great interest and scientific relevance. While I understand the space constraints inherent in contributions to OSR, I would encourage the authors to try to delve deeper into this analysis, as I believe it would enhance the scientific value of the contribution.

- **SPECIFIC COMMENTS**

L 10→ The temperature anomalies are an intrinsic part of MHWs; I understand what the author means but "thermal anomalies" per se, are not a PREcondition for MHWs, actually they are a condition.

L 35→ "In our BSH data": Reword the sentence avoiding familiarity with the data used.

L 41-49→ As is the case with the other databases, this paragraph does not explain the purpose of reanalysis data in this work.

L 52→ "The BACC Author Team (2008)": The word "The" is a definite article that, as far as I understand, does not take part in the name of the group. Thus, in the sentence, it must appear in lowercase, and the bibliographic reference must be "BACC Author Team" ordered under the letter 'B'.

L 80-81→ "rather continuously from 1989 until the present": Have MHWs been computed from incomplete datasets in this work? If so, discuss the implications.

L 90→ "1 nm": In scientific publications, units are expressed in accordance with the International System of Units (SI) or by using derived units as products of powers of these. Consequently, "1 nm" corresponds to one nanometre. I suggest expressing the spatial resolution in kilometres.

L 93→ Section 2.4 employs the detection of MHWs, which is explained later in Section 2.5. Therefore, the description of methods for detecting MHWs must precede the model validation.

L 103-104→ "In general though … station and model data": I think this is not well appreciated in the displayed figure. However, it would be interesting if, given that this is a study of MHWs, a direct comparison of the parameters that typically characterize MHWs is made: Number of events, duration, intensity, etc (see General Comments).

L 105-114→ Is it not possible to make a similar analysis for the deeper levels than the one performed in surface?

L 116→ MHWs were previously defined.

L 118-120→ Despite of both packages produce identical results; it is worth mentioning why the effort of using different packages. Is it because of the computational efficiency of Matlab package? If so, why not use Matlab package also for observational data?

L 126-129→ In the work, the methodology for computing MHWs is applied not only to product ref. no. 3 but also to product ref. no. 2 (Figure 1 and 4). Therefore, both products deserve the same treatment here. If there were differences between the climatologies used for both products, Section 2.4 must include a discussion of how these differences can affect the validation.

L 144→ SST anomaly rank is not a clear statistic. Please clarify.

L 154→ See L 10.

L 165→ "While the duration of …. Regions": It is not clear what region you refer.

L 175→ "total five": According to Fi. 4b there were four.

L 178-179→ "This trend of …. is taken into account": This discussion is confusing. Why it is necessary to include extra data to detect trends? I would include the full observational record in left panels of Figure 4 avoiding this discussion.

L 195→ "no temperature measurements exist in lower layers": As far as I see, this is the first time the reader knows this lack of data in the text. This must be stated much earlier, in sections 2.2 and 2.4.

L 221-226→ The information related to Holbrook et al. 2019 must be moved to Introduction.

L 232→ According to what I know, an exceedance of 9ºC could be one of the highest intensities observed of a MHW in the world. It might be interesting to look for some bibliographic reference to provide some context of this huge magnitude.

---

## Author Comment (AC1)

*The section addresses a relevant and very pertinent topic: the study of marine heatwaves in the Baltic Sea is extremely necessary. However, the section still requires more organization. The writing is often unclear. Important decisions were made for this section, although they still require support with the inclusion of more incisive justifications in the text. The section requires MINOR REVISIONS. Its current version cannot be approved for publication. To improve the manuscript, some questions are raised in the comments below, which can guide the authors.*

We thank the reviewer for their thoughtful and thorough review and helpful suggestions. We addressed the reviewer comments below in blue.

**General comments**

1) Why did the validation use only moorings (LT Kiel and Northern Baltic)? Are the analyses at two close stations (diamond and circle symbols in Fig. 2) representative for the entire Baltic Sea?

   In general, the model is validated with data from various stations distributed across the Baltic Sea. For the revised version of the section we will provide a more comprehensive model validation using k-means statistics, but move this to the supplementary material in agreement with the editor. Furthermore we will provide a map with all stations and place names mentioned in the text.

   In our Figure 2, actually three stations were marked - admittedly not too obviously. Two of those stations, 'Lighthouse Kiel' in the western Baltic Sea and 'Northern Baltic' in the north-east corner of the Baltic Proper, are used to validate the MHW parameters computed from the model (see Figure 4), since these two stations have relatively long records at the surface. Usually, the underlying climatology to assess MHW occurrences is computed from about 30 years worth of data or as close as possible to that.

   Please note that L82 currently states that the data at Northern Baltic is available down to depths of 103.8 m, which in fact is not the case. There is only surface data available. This will be corrected in the revised manuscript. The third station 'BMPH2' was therefore originally used to show the model's temperature skill at a deep station at 150m depth and close to Northern Baltic (see Figure 1c). While this monitoring station has data in greater depths, the availability is irregular in time and depth. So it is not possible to calculate a MHW climatology with this dataset.

   In the revised manuscript, we plan to drop the additional validation at BMPH2, as we do not actually use this station in the manuscript and the newly included k-means statistics will provide sufficient insight into the quality of the model over the entire Baltic Sea.

2) SST anomalies are computed with respect to a climatological SST considering the period 1997-2021, whereas MHWs are detected concerning the climatological data from 1993 to 2021. Why the choice of a dataset restricted to 1997-2021 for satellite SST? I suggest that the authors take a look at the link: https://data.marine.copernicus.eu/product/SST_BAL_SST_L4_REP_OBSERVATIONS_010_016/description

   Here, we wanted to use a dataset that is independent of the model data, i.e., our BSH operational routine, which - as mentioned in Sect. 2.1- is available after 1997. The

Copernicus Marine Baltic Sea SST reprocessed L3S product, which is what the L4 product is based on, is assimilated into the Baltic Sea reanalysis model and would therefore not represent an independent dataset.

3) In section 2.4 "Model validation", the author should describe the metrics and formulas applied to perform the analyses/study, and move Fig.1 and its description to a topic in section 3.
We will include Figs 1a and b as a supplement and drop BMPH2 (Fig 1c). As mentioned (see (1)), we will instead include a k-means model validation. The corresponding metrics and formulas will be included wherever appropriate.

4) OSR8 guideline suggests a maximum of 5 figures. Confirm it!
We will move the current Fig. 1 to the supplement. If permission is granted by the editor, we will instead add a map of the regions of the Baltic Sea as a new Fig. 1, which would still leave us with a maximum of 5 figures.

5) Improve the writing in general. For instance, Section 3.2.2 shows interesting results that can increase the impact of the section, but are not described carefully and clearly. Review it!
We will discuss Fig. 5 in more detail, especially regarding the subsurface MHWs and their potential drivers. We will also aim to improve the wording in the entire document.

6) English writing requires proofreading.
The revised manuscript will be proofread by an English native speaker.

**Specific comments**
(line 28) confirm the average period 1850-1900. Confirmed, the WMO State of the Global Climate 2022 provides this global mean anomaly relative to the 1850-1900 preindustrial reference period.

(line 29) fifth or sixth? Replace with "among the 6 warmest years" We will rephrase this accordingly in the revised manuscript.

(line 35) 1.35 K / 0.54°C; use same unit We will adjust to °C throughout the manuscript

(lines 51-70) Reduce the details to explain the satellite observations. Put their more relevant characteristics, then cite their main references in the text. - We will compress the section explaining the satellite observations, concentrating on the most relevant details.

(line 72) Remove "which are" will be removed

(line 75) replace the first "and" with "to".  will be replaced

(line 96) replace "chapter" with "section" will be replaced

(lines 99-107) This part describes results that should not be present in the section 2 "Data and Methods" As mentioned, Sect. 2.4 will be extensively modified

(lines 112-114) These lines describe results again. Organize the text and move them to the appropriate place. See above

(line 116) The acronym MHWs is already mentioned in line 25. will be adapted accordingly

(lines 118-120) If the tools produce identical results, it is not relevant to mention the use of python and Matlab. Mention only "We use open-source tools to detect MHW (Oliver, 2016; Zhao and Marin, 2019)" Will be changed

(line 126) "the climatological data of 1993 to 2022", replace "of" with "from". Figure 5 says 1993-2021.
The paragraph will be slightly rephrased. The climatological period will be homogenized to cover the years 1993 to 2021 for all MHW analyses based on the model data (Sect. 3.2, Sect. 3.2.2, Figs. 3 and 5).

(line 131) This line is not clear.
This paragraph has been reworked for clarification:
> *"Then, in order to evaluate the development of those MHW metrics over time, block averages (using a block length of one year) for each MHW metric are computed for both the observations (product ref. no 2 in Table 1) and the model data (product ref. no 3 in Table 1) at two stations: Lighthouse Kiel and Northern Baltic. The yearly MHW metrics from observations and the model are compared and the linear trends (95% significance) are calculated for each of those annual MHW metrics. Finally, the correlation of the annual MHW metrics to the annual mean temperature based on model data was assessed using a linear least-squares regression and a two-sided t-test for significance."*

(lines 138-140) "Bothnian Sea", "Bothnian Bay", "Baltic Proper", "Gulf of Finland" and "Gulf of Riga". Find a way to specify/highlight these regions in one of the maps in Fig. 2.
We will add an additional with a map of the regions of the Baltic Sea

(line 144) describe the metric "SST anomaly rank" in section 2.4 "Model validation"
The anomaly ranks shown in Fig. 2 are a very simple way to provide climatological context, providing information about how extreme an anomaly of a given magnitude is. For every grid point the anomalies of all years for a given month - e.g. August (calculated against the climatological mean of the respective month) - are sorted according to their magnitude. The anomaly rank depicts simply the rank in this sorting. For example, a plotted rank 1 of August 2022 in Fig. 2 therefore means that 2022 featured the hottest August in the entire dataset for the respective grid point. In Fig. 2 we show color shadings for the hottest eight and coldest eight ranks, respectively. We will provide a more detailed explanation in the revised manuscript.

(line 146) It would be better to avoid the mention of "obvious" here; replace it. Will be replaced

(lines 158-170) Specify the lat/lon points that delimiter the regions in discussion. The reading is difficult now. The regions will be depicted on an additional map of the Baltic Sea. Furthermore, the paragraph will be adjusted to improve the readability. We will also add a table (to the supplement?), which will contain the values of all regional MHW statistics. Furthermore, the climatology will be updated to cover only the years 1993 to 2021. Hence, the values of the statistics will be updated accordingly in the revised manuscript.

(lines 168-169) Would the Gulf of Bothnia include both the Bothnian Sea and Bothnian bay? Looking at Fig. 2, the highest SST anomalies are present in June and July in these regions. Why does the highest cumulative intensity derive from the temperature anomaly in November?

Yes, the Gulf of Bothnia includes the Bothnian Sea and the Bothnian Bay. Unfortunately, the statement in line 168-169 was incorrect. The high MHW metric values in the western Gulf of Bothnia stem mostly from a short, but very intense MHW that started at the end of June and ended at the beginning of July. This will be corrected in the revised manuscript.

The figures and metrics in Fig 2 and Fig 3 are not 100% comparable. The SST anomalies and the ranks are based on monthly mean values of weekly produced satellite-derived SST data within the data period 1997-2022. We use this independent dataset to complement the model and station data and to highlight the seasonal development of temperatures in the entire Baltic Sea.

The MHW analysis presented in Figure 3 is based on daily modeled SST data from the MYP covering the data period 1993-2022.

(lines 169-170) Sentence is also not clear. see above

(lines 176-177) replace "after 2020" with "in the evaluated period". will be replaced

(line 181) Replace "rising mean temperatures", my suggestion is "warming temperature trend"? Replaced with:

> "The maximum (Fig. 4c) and cumulative intensities (Fig. 4e) of observed MHWs do not show a clear trend and are not correlated to the warming annual mean temperatures (Fig. 4d and Fig. 4f)."

(lines 190-191) suggestion: replace "2018 was exceptional" with "In terms of total MHW days, the highest number is viewed in 2018" will be rephrased

(lines 203-204) "has a significant lower temperature of -0.3°C to 4.5°C as the climatological mean", this sentence is not clear, since Fig. 3b shows only 2022. Improve the writing Fig. 5b (which is what is referenced here) in fact shows the deviation between 2022 and the climatological mean. This will be clarified in the figure caption. We will also modify the sentence to make it clearer: "has a significantly lower temperature than the climatological mean (up to -7 °C deviation; Fig. 5b)"

(lines 208-209) "A few weeks prior to this MHW", is the author referring to the extreme or severe MHW? Specify the time period. Improve the writing Will be clarified and the paragraph extended to provide a more in-depth analysis of the (subsurface) MHWs.

(line 385) replace "chapter" with "section". will be replaced

(line 402) subplots 5g and 5h are not present will be removed from the caption

(Figure 3) change the color pallet The color pallet will be updated to be consistent with Fig. 2

(Figure 5) line 402 states "The period used for the climatology is 1993-2021", while line 126 says 1993-2022. In the end, what climatological period is used? The climatological period will be homogenized to 1993-2021 for all MHW analyses.

Improve the description of the caption in Fig. 5. will be done

---

## Author Comment (AC2)

Review of "Baltic Sea Surface Temperature Analysis 2022: A Study of Marine Heatwaves and Overall High Seasonal Temperatures" by Lindenthal et al. for the 8th edition of the Copernicus Marine Service Ocean State Report (OSR 8)

- SUMMARY

The present work conducts an analysis of the Marine Heatwaves (MHWs) detected in the Baltic Sea during the year 2022. To achieve this, the authors utilize various observational and modeling databases. The paper discusses the obtained results, focusing on the analysis of parameters characterizing the variability of MHWs throughout the year (intensity, frequency, and duration). The study describes the magnitude of MHWs in the Baltic environment and draws other interesting conclusions, such as the emergence of positive trends and the relationship between the vertical propagation of MHWs and the development of cold intermediate layers.
Hence, the preprint holds scientific value and falls within the scope of the Ocean State Report. However, some aspects requiring improvement have been identified for manuscript publication. These changes do not alter the substance of the work, allowing its publication with the implementation of minor revisions. Below is a list of the main reasons supporting this recommendation.
As a scientific reviewer, I am compelled to ensure the scientific quality of the contribution. Thus, despite being aware of the space limitations imposed on OSR submissions, some of the recommendations provided may conflict with these constraints. I leave it to the editor's discretion to decide on the implementation of such changes.

We thank the reviewer for their thoughtful and thorough review and helpful suggestions. We addressed the reviewer comments below in blue.

- GENERAL COMMENTS

1) The paper employs a considerable amount of geographical terminology (Bothnian Sea, Bothnian Bay, Baltic Proper, Gulf of Finland and Gulf of Riga).The use of this terminology is enriching and aids in the writing process; however, it should be noted that potential readers may not be familiar with Baltic geography. Therefore, I believe that the inclusion of an initial figure (map) displaying relevant data from the study area would be highly beneficial. This map could encompass the following information: Delimitation of the geographical zones used in the article, the location of observational stations (the markers in Figure 2 are hardly visible), bathymetry, etc.

   We will provide a new Figure with map and station and place names.

2) In the Introduction, I miss a clear statement of the objectives and motivation of this work.

   We will update the introduction and also the discussion in order to clarify our motivation and objectives in the revised manuscript.

3) One of the weakest points of the paper is related to the model validation (Section 2.4). While the validation exercise is appropriate,it is done inadequately in the paper.

   For the revised version of the section we will provide a more comprehensive model validation using k-means clustering algorithm (Raudsepp and Maljutenko, 2022), but move the details of this to the supplementary material in agreement with the editor. This method has been used for validation of the physical and biogeochemical models by Kõuts et al. (2022), Raudsepp et al. (2022, 2023).

Kõuts, M., Maljutenko, I., Elken, J., Liu, Y., Hansson, M., Viktorsson, L., Raudsepp, U., 2021. Recent regime of persistent hypoxia in the baltic sea. Environmental Research Communications, 3 (7), 075004. doi: 10.1088/2515-7620/ac0cc4

Raudsepp U, Maljutenko I. 2022. A method for assessment of the general circulation model quality using K-means clustering algorithm: a case study with GETM v2.5. Geosci Model Dev. 15:535–551. doi:10.5194/gmd-15-535-2022.

Raudsepp, U., Maljutenko, I., Haapala, J., Männik, A., Verjovkina, S., Uiboupin, R., von Schuckmann, K., Mayer, M., 2022. Record high heat content and lowice extent in the Baltic Sea during winter2019/20. In: Copernicus Ocean State Report, Issue 6, Journal of Operational Oceanography,15:sup1, s175–s185; DOI:10.1080/1755876X.2022.2095169

Raudsepp, U., Maljutenko, I., Barzandeh, A., Uiboupin, R., and Lagemaa, P., 2023. Baltic Sea freshwater content, in: 7th edition of the Copernicus Ocean State Report (OSR7), edited by: von Schuckmann, K., Moreira, L., Le Traon, P.-Y., Grégoire, M., Marcos, M., Staneva, J., Brasseur, P., Garric, G., Lionello, P., Karstensen, J., and Neukermans, G., Copernicus Publications, State Planet, 1-osr7, 7, https://doi.org/10.5194/sp-1-osr7-7-2023.

**Supplement**

We utilize a clustering method to assess the accuracy of the hydrodynamic model. This method provides insights into the overall model accuracy by clustering the errors. The clustering process employs the K-means algorithm, which is a form of unsupervised machine learning (Jain, 2010). The original description of this method can be found in the work of Raudsepp and Maljutenko (2022). In our assessment, all available data within the model domain and simulation period are included, even if the verification data is unevenly distributed or occasionally sparse. This approach allows us to evaluate the model quality at each specific location and time instance where measurements have been obtained.

The first step of the method is the formation of a two-dimensional error space of two simultaneously measured parameters. A two-dimensional error space ($dS,dT$) of simultaneously measured temperature and salinity values was formed as the basis for the clustering, where $dS=(S_{mod}-S_{obs})$ and $dT=(T_{mod}-T_{obs})$ with the model and observed salinity, $S_{mod}$ and $S_{obs}$, and the model and observed temperature, $T_{mod}$ and $T_{obs}$. The dataset utilized in this validation study was obtained from the EMODNET dataset compiled by SMHI (SMHI, 2019). It comprises a total of 3,094,089 observations that align with the simulation period of the Baltic Sea physics reanalysis (product ref. no. 3 in Table 1), encompassing the years 1993 to 2022. We extracted the nearest model values from the reanalysis dataset for each observation.

The second step is the selection of the number of clusters. For simplicity, we preselected five clusters. The third step is to perform a K-means clustering of the two-dimensional errors. The clustering is applied to the normalized errors. Normalization was done for temperature and salinity errors separately using corresponding standard deviations of the errors. The K-means algorithm finds the location of the centroids of a predefined number of clusters in the error space. The location of the centroids represents the bias of the set of errors for each cluster. The fourth step is calculation of statistical metrics of non-normalized clustered errors. Common statistics like standard deviation (STD), root mean square deviation (RMSD) and correlation coefficient can be calculated for the parameters belonging to each cluster.

The fifth step is the analysis of spatio-temporal distributions of the errors belonging to different clusters. In the formation of the error space, we retained the coordinates of each error point $(dS,dT)(x,y)$, which enables us to map the errors belonging to each cluster back to the location where the measurements were performed. In order to do that, the model domain is divided into horizontal grid cells $(i,j)$ of 27x27 km$^2$ size. Then the number of error points belonging to different clusters at each grid cell $(i,j)$ is counted. Total number of error points belonging to the grid cell $(i,j)$ is the sum of the points of each cluster. The share of error points in each grid cell belonging to cluster $k$ is the ratio of the number of error points of cluster $k$ and the total number of error points in each grid cell.

Figure A1 displays the results of the K-means clustering for non-normalized errors. Table S1 presents the corresponding metrics. Within cluster $k$=5, the salinity and temperature values closely align with the observations, with a bias of $dS$=-0.40 g/kg and $dT$=-0.02 °C, respectively. This cluster encompasses 57% of all data points. The points are distributed throughout the entire Baltic Sea, with a dominant share exceeding 0.5 (Figure S1b). Clusters $k$=3 and $k$=4 exhibit relatively even spatial distributions over the Baltic Sea, accounting for 11% and 8% of the points, respectively. These clusters are particularly noteworthy due to their relatively high temperature biases and variability, which are crucial for calculation of marine heatwaves. The clusters $k$=1 and $k$=2 represent the points with low temperature but high salinity error (Table S1). Spatially these points have high share in the southwestern Baltic Sea (Figure S1b) pointing to the occasional underestimation or overestimation of the inflow/outflow salinity.

Collectively, approximately 82% of all validation points exhibit relatively low temperature bias, STD, and RMSD (Table S1). The surface layer validation shows that less than 10% of comparison points have significant temperature errors (Figure S1c). Due to a low share of these validation points we do not expect the significant impact to the determination of the surface MHW and their statistics. Below the surface layer i.e., from a depth range of 0.5-40 m, the share of the clusters $k$=3 and $k$=4 could increase to 25% (Figure S1c). Consequently, we anticipate that the model reanalysis data provide sufficiently accurate information for calculating the subsurface MHW and their statistics of the Baltic Sea.

**References**

Jain, A. K. 2010. Data clustering: 50 years beyond K-means, Pattern Recognition Letters, 31, 651–666, doi:10.1016/j.patrec.2009.09.011

Raudsepp U, Maljutenko I. 2022. A method for assessment of the general circulation model quality using K-means clustering algorithm: a case study with GETM v2.5. Geosci Model Dev. 15:535–551. doi:10.5194/gmd-15-535-2022.

SMHI, 2019. Baltic Sea – Eutrophication and Acidity aggregated datasets 1902/2017 v2018, Aggregated datasets were generated in the framework of EMODnet Chemistry III, under the support of DG MARE Call for Tender EASME/EMFF/2016/006 – lot4, EMODnet Chemistry [data set], doi:10.6092/595D233C-3F8C-4497-8BD2-52725CEFF96B.

[Figure]

Figure S1. Distribution of normalized error clusters for K=5 (a). The spatial distribution (b, shaded sub-plots), vertical distribution (c), temporal distribution (d), and seasonal distribution (e) of the share of error points belonging to the five different clusters.

Table S1. The share (%), bias, root mean square deviation (RMSD), standard deviation (STD) and correlation coefficient (CORR) for each of the five clusters.

| k | share | BIAS dS | BIAS dT | STD dS | STD dT | RMSD S | RMSD T | CORR S | CORR T | CORR dSdT |
|---|---|---|---|---|---|---|---|---|---|---|
| 1 | 18 | -4.137 | -0.259 | 1.797 | 0.851 | 4.511 | 0.890 | 0.899 | 0.782 | -0.094 |
| 2 | 7 | 3.531 | 0.389 | 2.160 | 1.061 | 4.140 | 1.130 | 0.930 | 0.750 | -0.109 |
| 3 | 11 | -0.618 | 2.581 | 2.123 | 1.284 | 2.211 | 2.882 | 0.965 | 0.584 | -0.059 |
| 4 | 8 | 0.269 | -2.291 | 1.972 | 1.214 | 1.990 | 2.593 | 0.954 | 0.709 | -0.138 |
| 5 | 57 | -0.401 | -0.020 | 0.832 | 0.535 | 0.924 | 0.536 | 0.993 | 0.887 | 0.074 |

4) Additionally, the graphs used make it difficult to observe the drawn conclusions (see technical corrections). If the aim is to validate the model results regarding the detection of MHWs, I suggest that what should be validated is the model's estimation of the various parameters characterizing MHWs (i.e., the number of events, maximum intensity, cumulative intensity, and MHW days). This comparison can be easily conducted from the data already calculated in Figure 4 and presented to the reader using scatter plots and linear regressions.

We will amend section 3.2.1 with a statistical evaluation of the agreement between the computed MHW metrics based on the observations and the model. The table below shows the Pearson correlation coefficients for the MHW metrics in Fig 4 between observational and model data for the two stations.

Table #: Pearson correlation coefficients from linear regression between the MHW metrics computed from the station data and the model data at the stations Lighthouse Kiel and Northern Baltic.

| | common climatology period | MHW count | MH max intensity | MHW cumulative intensity | total MHW days |
|---|---|---|---|---|---|
| Lighthouse Kiel | 1993-2021 | 0.82 | 0.88 | 0.66 | 0.93 |
| Northern Baltic | 1997-2021 | 0.74 | 0.89 | 0.82 | 0.94 |

5) The analysis of the vertical structure of MHWs is of great interest and scientific relevance. While I understand the space constraints inherent in contributions to OSR, I would encourage the authors to try to delve deeper into this analysis, as I believe it would enhance the scientific value of the contribution.
We will discuss Fig. 5 in more detail, especially regarding the subsurface MHWs and their potential drivers.

- SPECIFIC COMMENTS

L 10->The temperature anomalies are an intrinsic part of MHWs; I understand what the author means but "thermal anomalies"per se, are not a PREcondition for MHWs, actually they are a condition. Will be changed to "condition"

L 35->"In our BSH data": Reword the sentence avoiding familiarity with the data used. Will be changed

L 41-49->As is the case with the other databases, this paragraph does not explain the purpose of reanalysis data in this work. Will be addressed

L 52->"The BACC Author Team (2008)": The word "The" is a definite article that, as far as I understand, does not take part in the name of the group. Thus, in the sentence, it must appear in lowercase, and the bibliographic reference must be "BACC Author Team" ordered under the letter 'B'. Will be changed

L 80-81->"rather continuously from 1989 until the present": Have MHWs been computed from incomplete datasets in this work? If so, discuss the implications.
Information on gaps in the observational data will be added to the text and uncertainties from this will be estimated based on Schlegel RW, Oliver ECJ, Hobday AJ and Smit AJ (2019) Detecting Marine Heatwaves With Sub-Optimal Data. *Front. Mar. Sci.* 6:737. doi: 10.3389/fmars.2019.00737

In the data from the two stations LT Kiel and Northern Baltic missing data has been quantified as follows:
LT Kiel             1993-2022: 8.6% of daily SST missing (9.1 % for 1989-2022)
Northern Baltic 1997-2022: 8.0 % of daily SST missing

According to Schlegel at al. (2019) for every percent point of missing data the degree of uncertainty introduced into the average marine heatwave (MHW) results is *on average* -0.44% in MHW count, -1.80% in duration and -0.31% in max. intensity.
Using a shorter time period than 30 years, e.g. here 1997-2022 for the station Northern Baltic, introduces an uncertainty of *on average* +0.32% per missing year on MHW count, -0.29% on duration and -0.23% on max. intensity. These average uncertainties can of course vary based on local conditions.

L 90->"1 nm": In scientific publications, units are expressed in accordance with the International System of Units (SI) or by using derived units as products of powers of these. Consequently, "1 nm"corresponds to one nanometre.I suggest expressing the spatial resolution in kilometres.
We will change the units to km

L 93->Section 2.4 employs the detection of MHWs, which is explained later in Section 2.5. Therefore, the description of methods for detecting MHWs must precede the model validation.
We can change the order

L 103-104->"In general though… station and model data": I think this is not well appreciated in the displayed figure. However, it would be interesting if, given that this Is a study of MHWs, a direct comparison of the parameters that typically characterize MHWs is made: Number of events, duration, intensity, etc. seeGeneral Comments). see point 4) above

L 105-114->Is it not possible to make a similar analysis for the deeper levels than the one performed in surface?
In terms of observational data we need stations with a long record of data. At Northern Baltic we only have SST and at Lighthouse Kiel the deepest datapoint is at 13 m, that is why we look at the model data in Fig 5 and section 3.2.2. We could only assess the deep 2022 MHW at Northern Baltic in a climatological context using model data. On the other hand, we do have limited space available for this report.
Please note in this context that there was a mistake in L. 82, where it was stated that the data at Northern Baltic was available down to depths of 103.8 m, which in fact is not the case. There is only surface data available. This will be corrected in the revised manuscript.

L 116->MHWs were previously defined. The definition will be removed

L 118-120->Despite of both packages produce identical results; it is worth mentioning why the effort of using different packages. Is it because of the computational efficiency of Matlab package? If so, why not use Matlab package also for observational data?
Out of the box, matlab can handle 2D data and we used it to produce 2D maps of 2022 MHW properties. The python code by default works with 1D data but also computes the block average for the climatological assessment of MHW.

L 126-129->In the work, the methodology for computing MHWs is applied not only to product ref.no. 3 but also to product ref. no. 2 (Figure 1 and 4). Therefore, both products deserve the same treatment here. If there were differences between the climatologies used for both products, Section 2.4 must include a discussion of how these differences can affect the validation.
We will harmonize which climatology period is used for all products and figures and will provide this in the revised manuscript. All MHW metrics will be computed based on the 1993-2021 climatology. Section 3.2 originally used 1993-2022.

L 144->SST anomaly rank is not a clear statistic. Please clarify.
The anomaly ranks shown in Fig. 2 are a very simple way to provide climatological context, providing information about how extreme an anomaly of a given magnitude is. For every grid point the anomalies of all years for a given month - e.g. August (calculated against the climatological mean of the respective month) - are sorted according to their magnitude. The anomaly rank depicts simply the rank in this sorting. For example, a plotted rank 1 of August 2022 in Fig. 2 therefore means that 2022 featured the hottest August in the entire dataset for the respective grid point. In Fig. 2 we show color shadings for the hottest eight and coldest eight ranks, respectively. We will provide a more detailed explanation in the revised manuscript.

L 154->See L 10. okay

L 165->"While the duration of .... Regions": It is not clear what region you refer.
The whole paragraph will be rephrased. We will also add an additional table which will help improve the readability of this section.

L 175->"total five": According to Fi. 4b there were four. A total of five in the observational data and four in the model data. This will be clarified in the revised manuscript.

L 178-179->"This trend of ....is taken into account": This discussion is confusing. Why it is necessary to include extra data to detect trends? I would include the full observational record in left panels of Figure 4 avoiding this discussion.
The full observational record is added to Fig 4, left, Station LT Kiel.

L 195->"no temperature measurements exist in lower layers": As far as I see, this is the first time the reader knows this lack of data in the text. This must be stated much earlier, in sections 2.2 and 2.4.
As mentioned, L. 82 falsely stated that there was measurement data available in lower levels at Northern Baltic. This will be corrected in the revised manuscript and we will clarify at this point that there are no long-term measurements in lower layers available that can be used to detect MHWs.

L 221-226->The information related to Holbrook et al. 2019 must be moved to Introduction. We will reorganize the discussion and move L. 217-226 to the introduction.

L 232->According to what I know, an exceedance of 9º C could be one of the highest intensities observed of a MHW in the world. It might be interesting to look for some bibliographic reference to provide some context of this huge magnitude.
As the MHW analysis will be redone using a climatological period from 1993 to 2021, this precise value may differ in the revised manuscript. Nevertheless, the maximum intensity here will remain rather high and we will discuss this in further detail, in particular with respect to local conditions (shallow water, sea ice, low seasonal temperature variations, etc.) that might lead to such enhanced local maximum intensities of MHWs.

---

## Author Response (AR1)

**Response to Editor and Reviewers**

We thank the editor and reviewers for their thoughtful and thorough reviews and helpful suggestions. In the revised manuscript, we have, in particular, made the following revisions:

- We have moved lines 217-226 from the discussion to the introduction and generally rewritten large parts of the introduction to clarify our motivation and objectives and justify the use of the individual datasets.
- We have moved the section "Model validation" to follow the section on "Heat wave detection". The revised section "Model validation" now highlights the relevant aspects for MHW detection from the corresponding Quality Information Document (QuID) and is further extended by an additional model validation using k-means statistics. The in-depth details of this additional approach are included as supplementary material (Appendix A1 in revised manuscript).
- The section "Results" contains several edits to enhance the readability of the manuscript:
  - We have added an overview map (new Figure 1) that introduces the various mentioned regions of the Baltic Sea
  - A new table (Table 3) now includes all MHW statistics for these regions and should help improve the readability of Sect. 3.2
- The details of Fig. 5 (vertical distribution of MHWs at the mooring station Northern Baltic) are now explained in greater detail in Sect. 3.2.2.
- The discussion has been extensively edited to further highlight the key points of the manuscript.

Regarding the figures (in particular, with respect to color schemes), we have made the following adjustments:

- The old Fig. 1 has been moved to the supplement (now Fig. A2) and no longer contains subfigure (c).
- Figure 1 is now a map of the Baltic Sea with relevant locations mentioned in the manuscript.
- Figure 2 no longer shows the locations of the used mooring stations (as these are now included in the new Fig. 1).
- The color maps of Figs. 3 and 5 e-f have been updated.
- Figure 4 has been updated, depicting a longer time series and using a red star for the year 2022.

Further, we address the individual editor and reviewer comments below in blue.

**Response to editor**

"I agree with your suggestion to add in the appendix a synthesis of the validation exercise that has been performed with the model and to refer to the published papers for details. A demonstration of the capabilities of the model to track MHW would be helpful to convince the reader. I would also encourage you to justify the choice of the period of computation for the climatology (1997-2021 instead of 1973[sic]-2021)."

We have included a more comprehensive model validation using k-means statistics (Sect. 2.5 and Appendix A1). Sect. 2.5 now also includes a statistical evaluation of the agreement

between the computed MHW metrics based on the observations and the model at two specific locations (lines 141-146 in revised manuscript), alongside a comparison of the annual temperature curves at these locations (lines 147-151 in revised manuscript).
The respective climatological periods are now explicitly specified and justified in Sect. 2.4 (lines 115-117 in revised manuscript).

**Response to Reviewer #1**

*The section addresses a relevant and very pertinent topic: the study of marine heatwaves in the Baltic Sea is extremely necessary. However, the section still requires more organization. The writing is often unclear. Important decisions were made for this section, although they still require support with the inclusion of more incisive justifications in the text. The section requires MINOR REVISIONS. Its current version cannot be approved for publication. To improve the manuscript, some questions are raised in the comments below, which can guide the authors.*

We thank the reviewer for their thoughtful and thorough review and helpful suggestions. We addressed the reviewer comments below in blue.

**General comments**
1) Why did the validation use only moorings (LT Kiel and Northern Baltic)? Are the analyses at two close stations (diamond and circle symbols in Fig. 2) representative for the entire Baltic Sea?
In general, the model is validated with data from various stations distributed across the Baltic Sea. For the revised version of the section we will provide a more comprehensive model validation using k-means statistics, but move this to the supplementary material in agreement with the editor. Furthermore we will provide a map with all stations and place names mentioned in the text.

In our Figure 2, actually three stations were marked - admittedly not too obviously. Two of those stations, 'Lighthouse Kiel' in the western Baltic Sea and 'Northern Baltic' in the north-east corner of the Baltic Proper, are used to validate the MHW parameters computed from the model (see Figure 4), since these two stations have relatively long records at the surface. Usually, the underlying climatology to assess MHW occurrences is computed from about 30 years worth of data or as close as possible to that.
Please note that L82 originally stated that the data at Northern Baltic is available down to depths of 103.8 m, which in fact is not the case. There is only surface data available. This has been corrected in the revised manuscript. The third station 'BMPH2' was originally used to show the model's temperature skill at a deep station at 150m depth and close to Northern Baltic (see previous Figure 1c). While this monitoring station has data in greater depths, the availability is irregular in time and depth. So it is not possible to calculate a MHW climatology with this dataset.
In the revised manuscript, we dropped the additional validation at BMPH2, as we do not actually use this station in the manuscript and the newly included k-means statistics provides sufficient insight into the quality of the model over the entire Baltic Sea.

2) SST anomalies are computed with respect to a climatological SST considering the period 1997-2021, whereas MHWs are detected concerning the climatological data from 1993 to 2021. Why the choice of a dataset restricted to 1997-2021 for satellite SST? I suggest that the authors take a look at the link: https://data.marine.copernicus.eu/product/SST_BAL_SST_L4_REP_OBSERVATIONS_0 10_016/description

   Here, we wanted to use a dataset that is independent of the model data, i.e., our BSH operational routine, which - as mentioned in Sect. 2.1- is available after 1997. The Copernicus Marine Baltic Sea SST reprocessed L3S product, which is what the L4 product is based on, is assimilated into the Baltic Sea reanalysis model and would therefore not represent an independent dataset.

3) In section 2.4 "Model validation", the author should describe the metrics and formulas applied to perform the analyses/study, and move Fig.1 and its description to a topic in section 3.
   The original Figs. 1a and b are now included as a supplement and BMPH2 (Fig 1c) has been dropped. As mentioned (see (1)), we have instead included a k-means model validation. The corresponding details of the approach can be found in Appendix A1.

4) OSR8 guideline suggests a maximum of 5 figures. Confirm it!
   We have moved the previous Fig. 1 to the supplement (now Fig. A2). As proposed, we have instead added a map of the regions of the Baltic Sea as a new Fig. 1, which still leaves us with a maximum of 5 figures.

5) Improve the writing in general. For instance, Section 3.2.2 shows interesting results that can increase the impact of the section, but are not described carefully and clearly. Review it!
   We have now included a more detailed discussion of Fig. 5, especially regarding the subsurface MHWs and their potential drivers. Further, we have checked and improved the wording in the entire document.

6) English writing requires proofreading.
   The revised manuscript has been proofread by an independent English native speaker.

**Specific comments**
(line 28) confirm the average period 1850-1900. Confirmed, the WMO State of the Global Climate 2022 provides this global mean anomaly relative to the 1850-1900 preindustrial reference period. This has been clarified in the manuscript.

(line 29) fifth or sixth? Replace with "among the 6 warmest years" This has been rephrased accordingly in the revised manuscript.

(line 35) 1.35 K / 0.54°C; use same unit We have adjusted to °C throughout the manuscript

(lines 51-70) Reduce the details to explain the satellite observations. Put their more relevant characteristics, then cite their main references in the text. - The section has been compressed, concentrating on the most relevant details.

(line 72) Remove "which are" has been removed

(line 75) replace the first "and" with "to".  has been replaced

(line 96) replace "chapter" with "section" has been replaced by "document"

(lines 99-107) This part describes results that should not be present in the section 2 "Data and Methods" As mentioned, Sect. 2.4 (now 2.5) has been extensively modified

(lines 112-114) These lines describe results again. Organize the text and move them to the appropriate place. See above

(line 116) The acronym MHWs is already mentioned in line 25. has been adapted accordingly

(lines 118-120) If the tools produce identical results, it is not relevant to mention the use of python and Matlab. Mention only "We use open-source tools to detect MHW (Oliver, 2016; Zhao and Marin, 2019)" Has been changed

(line 126) "the climatological data of 1993 to 2022", replace "of" with "from". Figure 5 says 1993-2021.
The paragraph has been rephrased. The climatological period has been homogenized to cover the years 1993 to 2021 for all MHW analyses based on the model data (Sect. 3.2, Sect. 3.2.2, Figs. 3 and 5).

(line 131) This line is not clear.
This paragraph has been reworked for clarification (cf. lines 109-114 in revised manuscript)

(lines 138-140) "Bothnian Sea", "Bothnian Bay", "Baltic Proper", "Gulf of Finland" and "Gulf of Riga". Find a way to specify/highlight these regions in one of the maps in Fig. 2.
We have added an additional figure with a map of the regions of the Baltic Sea

(line 144) describe the metric "SST anomaly rank" in section 2.4 "Model validation"
The anomaly ranks shown in Fig. 2 are a very simple way to provide climatological context, providing information about how extreme an anomaly of a given magnitude is. For every grid point the anomalies of all years for a given month - e.g. August (calculated against the climatological mean of the respective month) - are sorted according to their magnitude. The anomaly rank depicts simply the rank in this sorting. For example, a plotted rank 1 of August 2022 in Fig. 2 therefore means that 2022 featured the hottest August in the entire dataset for the respective grid point. In Fig. 2 we show color shadings for the hottest eight and coldest eight ranks, respectively. We have provided a more detailed explanation in the revised manuscript.

(line 146) It would be better to avoid the mention of "obvious" here; replace it. Has been replaced

(lines 158-170) Specify the lat/lon points that delimiter the regions in discussion. The reading is difficult now. The regions have been depicted on an additional map of the Baltic Sea. Furthermore, the paragraph has been edited to improve the readability. We have also added a table (Table 3), which contains the values of all regional MHW statistics. Furthermore, the climatology has been updated to cover only the years 1993 to 2021. However, as the Matlab tool that was used for the MHW analysis automatically omitted the considered year 2022

from the climatological period in the previous version of the manuscript, the values of the statistics still remained the same in the revised manuscript.

(lines 168-169) Would the Gulf of Bothnia include both the Bothnian Sea and Bothnian bay? Looking at Fig. 2, the highest SST anomalies are present in June and July in these regions. Why does the highest cumulative intensity derive from the temperature anomaly in November?

Yes, the Gulf of Bothnia includes the Bothnian Sea and the Bothnian Bay. Unfortunately, the statement in line 168-169 was incorrect. The high MHW metric values in the western Gulf of Bothnia stem mostly from a short, but very intense MHW that started at the end of June and ended at the beginning of July. This has been corrected in the revised manuscript.

The figures and metrics in Fig 2 and Fig 3 are not 100% comparable. The SST anomalies and the ranks are based on monthly mean values of weekly produced satellite-derived SST data within the data period 1997-2022. We use this independent dataset to complement the model and station data and to highlight the seasonal development of temperatures in the entire Baltic Sea.

The MHW analysis presented in Figure 3 is based on daily modeled SST data from the MYP covering the data period 1993-2022.

(lines 169-170) Sentence is also not clear. see above

(lines 176-177) replace "after 2020" with "in the evaluated period". has been modified

(line 181) Replace "rising mean temperatures", my suggestion is "warming temperature trend"? Replaced with: *"the warming annual mean temperatures."*

(lines 190-191) suggestion: replace "2018 was exceptional" with "In terms of total MHW days, the highest number is viewed in 2018" has been replaced with: "2018 shows the highest numbers"

(lines 203-204) "has a significant lower temperature of -0.3°C to 4.5°C as the climatological mean", this sentence is not clear, since Fig. 3b shows only 2022. Improve the writing Fig. 5b (which is what is referenced here) in fact shows the deviation between 2022 and the climatological mean. This has been clarified in the figure caption. We have also modified the sentence to make it clearer (lines 218-220 in revised manuscript)

(lines 208-209) "A few weeks prior to this MHW", is the author referring to the extreme or severe MHW? Specify the time period. Improve the writing Has been clarified and the paragraph extended to provide a more in-depth analysis of the (subsurface) MHWs.

(line 385) replace "chapter" with "section". Sentence omitted in revised manuscript

(line 402) subplots 5g and 5h are not present has been removed from the caption

(Figure 3) change the color pallet Has been changed

(Figure 5) line 402 states "The period used for the climatology is 1993-2021", while line 126 says 1993-2022. In the end, what climatological period is used? The climatological period has been homogenized to 1993-2021 for all MHW analyses based on model data.

Improve the description of the caption in Fig. 5. has been done

**Response to Reviewers #2**

Review of "Baltic Sea Surface Temperature Analysis 2022: A Study of Marine Heatwaves and Overall High Seasonal Temperatures" by Lindenthal et al. for the 8th edition of the Copernicus Marine Service Ocean State Report (OSR 8)

- **SUMMARY**

The present work conducts an analysis of the Marine Heatwaves (MHWs) detected in the Baltic Sea during the year 2022. To achieve this, the authors utilize various observational and modeling databases. The paper discusses the obtained results, focusing on the analysis of parameters characterizing the variability of MHWs throughout the year (intensity, frequency, and duration). The study describes the magnitude of MHWs in the Baltic environment and draws other interesting conclusions, such as the emergence of positive trends and the relationship between the vertical propagation of MHWs and the development of cold intermediate layers.
Hence, the preprint holds scientific value and falls within the scope of the Ocean State Report. However, some aspects requiring improvement have been identified for manuscript publication. These changes do not alter the substance of the work, allowing its publication with the implementation of minor revisions. Below is a list of the main reasons supporting this recommendation.
As a scientific reviewer, I am compelled to ensure the scientific quality of the contribution. Thus, despite being aware of the space limitations imposed on OSR submissions, some of the recommendations provided may conflict with these constraints. I leave it to the editor's discretion to decide on the implementation of such changes.

We thank the reviewer for their thoughtful and thorough review and helpful suggestions. We
addressed the reviewer comments below in blue.

- **GENERAL COMMENTS**

  1) The paper employs a considerable amount of geographical terminology (Bothnian Sea, Bothnian Bay, Baltic Proper, Gulf of Finland and Gulf of Riga).The use of this terminology is enriching and aids in the writing process; however, it should be noted that potential readers may not be familiar with Baltic geography. Therefore, I believe that the inclusion of an initial figure (map) displaying relevant data from the study area would be highly beneficial. This map could encompass the following information: Delimitation of the geographical zones used in the article, the location of observational stations (the markers in Figure 2 are hardly visible), bathymetry, etc.

     We have provided a new Figure with a map and station and place names (new Fig. 1).

2) In the Introduction, I miss a clear statement of the objectives and motivation of this work.

We have updated the introduction and also the discussion in order to clarify our motivation and objectives in the revised manuscript.

3) One of the weakest points of the paper is related to the model validation (Section 2.4). While the validation exercise is appropriate,it is done inadequately in the paper.

For the revised version of the section we have provided a more comprehensive model validation using a k-means clustering algorithm (Raudsepp and Maljutenko, 2022), but moved the details of this to the supplementary material in agreement with the editor (Appendix A1, Fig. A1, Table A1). This method has been used for validation of the physical and biogeochemical models by Kõuts et al. (2022), Raudsepp et al. (2022, 2023).

Kõuts, M., Maljutenko, I., Elken, J., Liu, Y., Hansson, M., Viktorsson, L., Raudsepp, U., 2021. Recent regime of persistent hypoxia in the baltic sea. Environmental Research Communications, 3 (7), 075004. doi: 10.1088/2515-7620/ac0cc4

Raudsepp U, Maljutenko I. 2022. A method for assessment of the general circulation model quality using K-means clustering algorithm: a case study with GETM v2.5. Geosci Model Dev. 15:535–551. doi:10.5194/gmd-15-535-2022.

Raudsepp, U., Maljutenko, I., Haapala, J., Männik, A., Verjovkina, S., Uiboupin, R., von Schuckmann, K., Mayer, M., 2022. Record high heat content and lowice extent in the Baltic Sea during winter2019/20. In: Copernicus Ocean State Report, Issue 6, Journal of Operational Oceanography,15:sup1, s175–s185; DOI:10.1080/1755876X.2022.2095169

Raudsepp, U., Maljutenko, I., Barzandeh, A., Uiboupin, R., and Lagemaa, P., 2023. Baltic Sea freshwater content, in: 7th edition of the Copernicus Ocean State Report (OSR7), edited by: von Schuckmann, K., Moreira, L., Le Traon, P.-Y., Grégoire, M., Marcos, M., Staneva, J., Brasseur, P., Garric, G., Lionello, P., Karstensen, J., and Neukermans, G., Copernicus Publications, State Planet, 1-osr7, 7, https://doi.org/10.5194/sp-1-osr7-7-2023.

4) Additionally, the graphs used make it difficult to observe the drawn conclusions (see technical corrections). If the aim is to validate the model results regarding the detection of MHWs, I suggest that what should be validated is the model's estimation of the various parameters characterizing MHWs (i.e., the number of events, maximum intensity, cumulative intensity, and MHW days). This comparison can be easily conducted from the data already calculated in Figure 4 and presented to the reader using scatter plots and linear regressions.

We have amended the model validation (Sect. 2.5) with a statistical evaluation of the

agreement between the computed MHW metrics based on the observations and the model (lines 141-146 in revised manuscript), alongside a comparison of the annual temperature curves at these locations (lines 147-151) in revised manuscript

5) The analysis of the vertical structure of MHWs is of great interest and scientific relevance. While I understand the space constraints inherent in contributions to OSR, I would encourage the authors to try to delve deeper into this analysis, as I believe it would enhance the scientific value of the contribution.
We have included a more in-depth discussion of Fig. 5, especially regarding the subsurface MHWs and their potential drivers.

- **SPECIFIC COMMENTS**

L 10->The temperature anomalies are an intrinsic part of MHWs; I understand what the author means but "thermal anomalies"per se, are not a PREcondition for MHWs, actually they are a condition. Has been changed accordingly

L 35->"In our BSH data": Reword the sentence avoiding familiarity with the data used. Has been changed

L 41-49->As is the case with the other databases, this paragraph does not explain the purpose of reanalysis data in this work. Lines 48-53 in the revised manuscript justify the use of the respective datasets

L 52->"The BACC Author Team (2008)": The word "The" is a definite article that, as far as I understand, does not take part in the name of the group. Thus, in the sentence, it must appear in lowercase, and the bibliographic reference must be "BACC Author Team" ordered under the letter 'B'. Has been changed

L 80-81->"rather continuously from 1989 until the present": Have MHWs been computed from incomplete datasets in this work? If so, discuss the implications.
Information on gaps in the observational data has been added to the text (lines 82 and 84). Uncertainties from this can be estimated based on Schlegel RW, Oliver ECJ, Hobday AJ and Smit AJ (2019) Detecting Marine Heatwaves With Sub-Optimal Data. *Front. Mar. Sci.* 6:737. doi: 10.3389/fmars.2019.00737

In the data from the two stations LT Kiel and Northern Baltic missing data has been quantified as follows:
LT Kiel        1993-2022: 8.6% of daily SST missing (9.1 % for 1989-2022)
Northern Baltic 1997-2022: 8.0 % of daily SST missing

According to Schlegel at al. (2019) for every percent point of missing data the degree of uncertainty introduced into the average marine heatwave (MHW) results is *on average* -0.44% in MHW count, -1.80% in duration and -0.31% in max. intensity.
Using a shorter time period than 30 years, e.g. here 1997-2022 for the station Northern Baltic, introduces an uncertainty of *on average* +0.32% per missing year on MHW count, -0.29% on duration and -0.23% on max. intensity. These average uncertainties can of course vary based on local conditions.

L 90->"1 nm": In scientific publications, units are expressed in accordance with the International System of Units (SI) or by using derived units as products of powers of these. Consequently, "1 nm"corresponds to one nanometre.I suggest expressing the spatial resolution in kilometres.
Has been modified accordingly

L 93->Section 2.4 employs the detection of MHWs, which is explained later in Section 2.5. Therefore, the description of methods for detecting MHWs must precede the model validation.
The order of Sections 2.4 and 2.5 has been changed

L 103-104->"In general though... station and model data": I think this is not well appreciated in the displayed figure. However, it would be interesting if, given that this Is a study of MHWs, a direct comparison of the parameters that typically characterize MHWs is made: Number of events, duration, intensity, etc. seeGeneral Comments). see point 4) above

L 105-114->Is it not possible to make a similar analysis for the deeper levels than the one performed in surface?
In terms of observational data we need stations with a long record of data. At Northern Baltic we only have SST and at Lighthouse Kiel the deepest datapoint is at 13 m. That is why we look at the model data in Fig. 5 and Sect. 3.2.2. We could only assess the deep 2022 MHW at Northern Baltic in a climatological context using model data.
Please note in this context that there was a mistake in L. 82 of the original manuscript, where it was stated that the data at Northern Baltic was available down to depths of 103.8 m, which in fact is not the case. There is only surface data available. This has been corrected in the revised manuscript.

L 116->MHWs were previously defined. Has been changed

L 118-120->Despite of both packages produce identical results; it is worth mentioning why the effort of using different packages. Is it because of the computational efficiency of Matlab package? If so, why not use Matlab package also for observational data?
Out of the box, matlab can handle 2D data and we used it to produce 2D maps of 2022 MHW properties. The python code by default works with 1D data but also computes the block average for the climatological assessment of MHW. Lines 99-100 of the revised manuscript no longer explicitly mention the two different open-source tools.

L 126-129->In the work, the methodology for computing MHWs is applied not only to product ref.no. 3 but also to product ref. no. 2 (Figure 1 and 4). Therefore, both products deserve the same treatment here. If there were differences between the climatologies used for both products, Section 2.4 must include a discussion of how these differences can affect the validation.
We have harmonized which climatology period is used for all products and figures and provided this information in the revised manuscript (lines 115-117 in revised manuscript). Furthermore lines 105-114 now clearly state which methods were applied to which dataset.

L 144->SST anomaly rank is not a clear statistic. Please clarify.
The anomaly ranks shown in Fig. 2 are a very simple way to provide climatological context, providing information about how extreme an anomaly of a given magnitude is. For every grid point the anomalies of all years for a given month - e.g. August (calculated against the

climatological mean of the respective month) - are sorted according to their magnitude. The anomaly rank depicts simply the rank in this sorting. For example, a plotted rank 1 of August 2022 in Fig. 2 therefore means that 2022 featured the hottest August in the entire dataset for the respective grid point. In Fig. 2 we show color shadings for the hottest eight and coldest eight ranks, respectively. We have provided a more detailed explanation in the revised manuscript.

L 154->See L 10. okay

L 165->"While the duration of …. Regions": It is not clear what region you refer.
The whole paragraph has been rephrased. We have also added an additional table (Table 3) which should help improve the readability of this section.

L 175->"total five": According to Fi. 4b there were four. A total of five in the observational data and four in the model data. This has been clarified in the revised manuscript.

L 178-179->"This trend of ….is taken into account": This discussion is confusing. Why it is necessary to include extra data to detect trends? I would include the full observational record in left panels of Figure 4 avoiding this discussion.
The full observational record has been added to Fig 4, left, Station LT Kiel.

L 195->"no temperature measurements exist in lower layers": As far as I see, this is the first time the reader knows this lack of data in the text. This must be stated much earlier, in sections 2.2 and 2.4.
As mentioned, L. 82 of the original manuscript falsely stated that there was measurement data available in lower levels at Northern Baltic. This has been corrected in the revised manuscript and we have clarified at this point that there are no long-term measurements in lower layers available that can be used to detect MHWs (lines 85-86 in revised manuscript).

L 221-226->The information related to Holbrook et al. 2019 must be moved to Introduction.
We have reorganized the discussion and moved L. 217-226 to the introduction.

L 232->According to what I know, an exceedance of 9º C could be one of the highest intensities observed of a MHW in the world. It might be interesting to look for some bibliographic reference to provide some context of this huge magnitude.
We have included a discussion of this high maximum intensity in Sect. 4 (lines 247-253 in revised manuscript).

---

## Referee Report (RR1)

The section addresses a relevant and very pertinent topic: the study of marine heatwaves in the Baltic Sea is extremely necessary. The authors have improved the manuscript in relation to its first submitted version, but it still requires MINOR REVISIONS to be accepted. There are a few comments below which can guide the authors to improve the overall manuscript.

**General comment:**

Sometimes the authors refer the reanalysis results purely as "model data/results". "Model" is very general. My suggestion is to always use "reanalysis" and standardize along the text.

**Specific comments:**

(line 61) What is the reason to cite BACC Author Team (2008) and Gröger et al. (2022)? What did they study?

(line 81) replace "Of" with "Regarding".

(line 84) replace "… Baltic Proper. The SST observations…" with "… Baltic Proper where the SST observations…"

(line 88) You can also specify Baltic Sea physics reanalysis product with an acronym.

(lines 102-104) Cite the corresponding MHW reference which defines the MHW categories.

(line 106) "MYP" is very general. My suggestion is to refer to the Baltic Sea physics reanalysis here (and along the manuscript) using a proper acronym for the Baltic Sea reanalysis as it was recommended in line 88. Multi-year product (MYP) is a specific nomenclature used mostly by Copernicus Marine Service to define different products such as an ocean reanalysis.

(line 106-107) Move "the following statistical metrics" to the end of the sentence as follows: "MHWs are computed at every third surface grid point of the MYP, resulting in a resolution of approximately 5.4 km for the following statistical metrics: …"

(lines 119-120) Move the following sentence to the section 2.3: "The MYP data has already been extensively validated in the corresponding Quality Information Document (QuID; Panteleit et al., 2023)".

(lines 121-124) Again, my suggestion is to move these results to an appropriate section in "Results".

(lines 220-222) "This also coincides with the onset of significantly higher temperatures at the surface compared to the climatological mean, though these were initially not high enough to result in a MHW (Fig. 5e)". Figure 5e shows two MHW events at 0.5 m, right? Rewrite or clarify.

(lines 222-226) Specify these lines describe the results at 0.5 m (Figure 5e).

(lines 208-238) the following subsection "3.2.2 Analysis of vertical MHW distribution at Northern Baltic" still requires writing improvements.

(line 278) Appendix A1 is not well connected with the text. Clearly explain why the analysis and results in Appendix A1 are important.

(line 279) "hydrodynamic model". Reanalysis?

(line 496) replace "model data" with "reanalysis data". See the above general comment.

---

## Author Response (AR2)

**Response to Reviewers**

We thank the reviewers for their reports and, in particular, reviewer #2 for their helpful suggestions. In the revised manuscript, we have addressed reviewer #2's comments in the following manner:

**General comment:**
Sometimes the authors refer the reanalysis results purely as "model data/results". "Model" is very general. My suggestion is to always use "reanalysis" and standardize along the text.
- We have replaced "model data" by "reanalysis data" and have generally homogenized the manuscript in this respect. The Baltic Sea physics reanalysis product is now either abbreviated with the newly introduced acronym (BAL-MYP) or simply addressed with "reanalysis", wherever it is clear that the BAL-MYP is meant.

**Specific comments:**
(line 61) What is the reason to cite BACC Author Team (2008) and Gröger et al. (2022)? What did they study?
- We have now specified this in lines 60-62.

(line 81) replace "Of" with "Regarding".
- done

(line 84) replace "… Baltic Proper. The SST observations…" with "… Baltic Proper where the SST observations…"
- done

(line 88) You can also specify Baltic Sea physics reanalysis product with an acronym.
- done: BAL-MYP (introduced in line 89, added to Table 1)

(lines 102-104) Cite the corresponding MHW reference which defines the MHW categories.
- The citation is already mentioned in line 101. We have added a colon to clarify that lines 102-104 describe this classification of MHW categories.

(line 106) "MYP" is very general. My suggestion is to refer to the Baltic Sea physics reanalysis here (and along the manuscript) using a proper acronym for the Baltic Sea reanalysis as it was recommended in line 88. Multi-year product (MYP) is a specific nomenclature used mostly by Copernicus Marine Service to define different products such as an ocean reanalysis.
- done (as mentioned above)

(line 106-107) Move "the following statistical metrics" to the end of the sentence as follows: "MHWs are computed at every third surface grid point of the MYP, resulting in a resolution of approximately 5.4 km for the following statistical metrics: …"
- done

(lines 119-120) Move the following sentence to the section 2.3: "The MYP data has already been extensively validated in the corresponding Quality Information Document (QuID; Panteleit et al., 2023)".
- As section 2.5 is specifically dedicated to the validation of the BAL-MYP with respect to its ability to accurately quantify MHWs, we would prefer to leave this sentence here. Furthermore, the following lines (lines 121-125) present selected results from the QuID that are relevant for this study.

(lines 121-124) Again, my suggestion is to move these results to an appropriate section in "Results".
- These lines do not represent results made in our study. We have rewritten the text to clarify that these results stem from the QuID.

(lines 220-222) "This also coincides with the onset of significantly higher temperatures at the surface compared to the climatological mean, though these were initially not high enough to result in a MHW (Fig. 5e)". Figure 5e shows two MHW events at 0.5 m, right? Rewrite or clarify.

- We have rewritten the section to clarify that the results near the surface are based on reanalysis data at 0.5 m depth.

(lines 222-226) Specify these lines describe the results at 0.5 m (Figure 5e).

- done, see above

(lines 208-238) the following subsection "3.2.2 Analysis of vertical MHW distribution at Northern Baltic" still requires writing improvements.

- We have improved the overall wording in this section

(line 278) Appendix A1 is not well connected with the text. Clearly explain why the analysis and results in Appendix A1 are important.

- Appendix A1 is referenced and summarized in the main article (lines 126-141), where it is stated that this clustering approach was used as an additional validation of the BAL-MYP, in order to highlight its ability to accurately capture both surface and subsurface MHWs over the entire domain. We have now clarified this at the beginning of Appendix A1 in lines 281-282.

(line 279) "hydrodynamic model". Reanalysis?

- Yes, this has been rewritten accordingly

(line 496) replace "model data" with "reanalysis data". See the above general comment.

- done